# *Fem-1* Gene of Chinese White Pine Beetle (*Dendroctonus armandi*): Function and Response to Environmental Treatments

**DOI:** 10.3390/ijms251910349

**Published:** 2024-09-26

**Authors:** Jiajin Wang, Songkai Liao, Haoyu Lin, Hongjian Wei, Xinjie Mao, Qi Wang, Hui Chen

**Affiliations:** 1State Key Laboratory of Conservation and Utilization of Subtropical Agro-Bioresources, Guangdong Laboratory for Lingnan Modern Agriculture, College of Forestry and Landscape Architecture, South China Agricultural University, Guangzhou 510462, China; a947375051@stu.scau.edu.cn (J.W.); liaosongkai2023@163.com (S.L.); weihongjian@stu.scau.edu.cn (H.W.); mxj1113@outlook.com (X.M.); wangqi07250@163.com (Q.W.); 2Forest Protection Research Institute, Fujian Academy of Forestry, Fuzhou 350011, China; jolhy@163.com

**Keywords:** *Dendroctonus armandi*, *fem-1*, sex determination, environments

## Abstract

*Dendroctonus armandi* (Tsai and Li) (Coleoptera: Curculionidae: Scolytinae) is regarded as the most destructive forest pest in the Qinling and Bashan Mountains of China. The sex determination of *Dendroctonus armandi* plays a significant role in the reproduction of its population. In recent years, the role of the *fem-1* gene in sex determination in other insects has been reported. However, the function and expression of the *fem-1* gene in *Dendroctonus armandi* remain uncertain. In this study, three *fem-1* genes were cloned and characterized. These were named *Dafem-1*A, *Dafem-1*B, and *Dafem-1*C, respectively. The expression levels of these three *Dafem-1* genes vary at different stages of development and between the sexes. In response to different environmental treatments, including temperature, nutrients, terpenoids, and feeding duration, significant differences were observed between the three *Dafem-1* genes at different developmental stages and between males and females. Furthermore, injection of double-stranded RNA (dsRNA) targeting the expressions of the *Dafem-1*A, *Dafem-1*B, and *Dafem-1*C genes resulted in increased mortality, deformity, and decreased emergence rates, as well as an imbalance in the sex ratio. Following the interference with *Dafem-1*A and *Dafem-1*C, no notable difference was observed in the expression of the *Dafem-1*B gene. Similarly, after the interference with the *Dafem-1*B gene, no significant difference was evident in the expression levels of the *Dafem-1*A and *Dafem-1*C genes. However, the interference of either the *Dafem-1*A or *Dafem-1*C gene results in the downregulation of the other gene. The aforementioned results demonstrate that the *Dafem-1*A, *Dafem-1*B, and *Dafem-1*C genes play a pivotal role in the regulation of life development and sex determination. Furthermore, it can be concluded that external factors such as temperature, nutrition, terpenoids, and feeding have a significant impact on the expression levels of the *Dafem-1*A, *Dafem-1*B, and *Dafem-1*C genes. This provides a crucial theoretical foundation for further elucidating the sex determination mechanism of *Dendroctonus armandi*.

## 1. Introduction

The Chinese white pine beetle (*Dendroctonus armandi* Tsai and Li) is a highly destructive pioneer pest in the Qinling and Bashan Mountains forest area of China [1,2]. It invades the bark of the Chinese white pine (*Pinus armandii* France) and corrodes the phloem, resulting in a significant decline in the vigor of *P. armandii* [3,4,5]. Female beetles initially colonize the tree, subsequently releasing sex pheromones to attract males for mating and colonization, thereby increasing the population size [6]. Thus, elucidating the mechanism of sex development in *D. armandi* offers a theoretical foundation for the population management of *D. armandi*. However, research on sex determination in *D. armandi* is still in its early stages, and the functions and expressions of many sex-determining genes urgently need to be explored.

The *feminization*-1 (*fem-1*) gene encodes an intracellular protein with a conserved anchor repeat motif that mediates protein–protein interactions [7,8]. The *fem-1* gene was first identified in the nematode *Caenorhabditis elegans* and is a key regulatory gene involved in male and hermaphroditic spermatogenesis [9]. The *fem-1* gene family comprises three members: *fem-1*A, *fem-1*B, and *fem-1*C. These genes typically share a unique and conserved anchor protein repeat domain [10]. A considerable number of *fem-1* homologous genes have been identified and characterized across a diverse range of species, including mammals, invertebrates, and insects. Examples include humans, mice, zebrafish, and the *Manila migratory* locust [11,12]. It is postulated that they should have different biological activities and evolutionary processes in order to respond to different physiological functions [13]. In recent years, the *fem-1* gene has been cloned in crustaceans and found to have a bisexual gene expression pattern. For instance, Song et al. described the characteristics of three members of the *fem-1* gene family in the crab species *Eriocheir sinensis* and demonstrated that *Esfem-1* may play a role in early sex determination and late gonadal development in crabs [14]. Similarly, the *fem-1* gene has been identified in the oriental river pawn, *Macrobrachium nipponense,* and *Macrobrachium rosenbergii*. The research results for *M. nipponense* demonstrated that the *fem-1* gene is highly expressed in the ovaries of female specimens. In situ hybridization results indicated that the *fem-1* gene exhibits a strong positive signal concentrated at the edge of oocytes prior to yolk formation [15,16]. The research results for *M. rosenbergii* demonstrated that the *fem-1* gene is exclusively expressed in the ovary. Furthermore, the expression level of the *fem-1* gene was observed to increase with ovarian maturation, reaching its peak during yolk formation [17]. This evidence suggests that the *fem-1* gene may play a role in ovarian development and maturation. The *fem-1* gene was also cloned in other species, including *Mus musculus*, *Taeniopygia guttata*, *Pinctada margaritifera*, *Chlamys nobilis*, *Charybdis feriatus*, *Sipunculus nudus*, *Gigantidas haimaensis*, *Fenneropenaeus merguiensis*, and so on [18,19,20,21,22,23,24,25].

RNA interference (RNAi) is the process by which complementary endogenous messenger RNA (mRNA) is silenced by exogenous double-stranded RNA (dsRNA) [26]. Following the initial discovery of RNAi in *C. elegans* [27], the first successful application of this technique in insects was in the fruit fly *Drosophila melanogaster* [28]. Given its high degree of specificity, RNAi represents a promising approach to pest management [29,30,31]. It has rapidly evolved into a tool that is now widely used in a variety of insects, including those belonging to Hymenoptera [32,33], Diptera [34,35], Coleoptera [36,37], and Lepidoptera [38,39,40]. Two methods of delivering dsRNA to insect target tissues have been developed: injection and ingestion [41,42]. RNAi technology is widely used in *D. armandi* research (including this study) and has yielded many results in studying various gene functions [2]. In addition to the aforementioned *C. elegans*, RNAi has also been employed for the study of *fem-1* genes in other species, including *Cherax quadricarinatus* (redclaw crayfish) and *E. sinensis*. Following the RNAi of the *fem-1*B gene, a notable reduction in the expression of the *VTG* gene was observed in the ovaries and liver pancreas of *C. quadricarinatus* [43]. Similarly, in *E. sinensis*, RNA interference with the *fem-1*C gene resulted in a considerable decline in the expression level of the *Esfem-1*C gene, with notable implications for male development [14].

Nevertheless, the role of the *fem-1* gene in the sex determination of *D. armandi* remains uncertain, and there are few reports on the *fem-1* gene. *D. armandi* has two generations per year at altitudes below 1700 m, three generations per two years at altitudes between 1750 and 2100 m, and only one generation per year at altitudes above 2100 m [44]. Furthermore, in the Qinling and Bashan Mountains at 1700m, *D. armandi* exhibits sexual dimorphism [1], with females in excess of males and females generally having larger body sizes than males [45]. Combining the role of the *fem-1* gene in the sex determination of *C. elegans*, this study not only investigates the role of the *fem-1* gene in the sex determination of *D. armandi* but also examines the effects of temperature, nutrition, feeding, terpenoids, and other treatments to reduce environmental factors on the sex-determining genes of *D. armandi*, thus better revealing the underlying reasons for the sexual dimorphism of *D. armandi*. RNA interference was employed to silence the three *Dafem-1* genes, thereby enabling the study of its gene function. This method offers a theoretical foundation for the study of sex determination in *D. armandi* and the utilization of RNAi in the management of *D. armandi* populations.

## 2. Results

### 2.1. Sequencing Analysis of Dafem-1

In this experiment, three *Dafem-1* genes were cloned and obtained. The results of the gel electrophoresis of some target fragments are presented in Appendix A. These genes have been uploaded to the NCBI database. The *fem-1* genes are named *Dafem-1*A (PP738886), *Dafem-1*B (PP738887), and *Dafem-1*C (PP738888), with sequence lengths of 2180 bp, 2307 bp, and 2173 bp, respectively (Appendix A). The *Dafem-1*A, *Dafem-1*B, and *Dafem-1*C genes encode 630, 662, and 650 amino acids, respectively, with molecular weights of 17.86, 18.92, and 17.84, and theoretical isoelectric points of 4.92, 4.90, and 4.92, respectively (Table 1). A sequence percentage identity analysis was conducted on the nucleotide and protein sequences of the three *Dafem-1* genes, the results of which demonstrated that the nucleotide sequence percentage identity between the *Dafem-1*A and *Dafem-1*B genes is 49.17%, while the nucleotide sequence percentage identity with the *Dafem-1*C gene is 45.31%. The nucleotide sequence percentage identity between the *Dafem-1*B and *Dafem-1*C genes is 43.04%. The protein sequence percentage identity between the *Dafem-1*A and *Dafem-1*B genes is 38.13%, and the protein sequence percentage identity with the *Dafem-1*C gene is 31.87%. The protein sequence percentage identity between the *Dafem-1*B and *Dafem-1*C genes is 26.09% (Table 2). Following the sequence analysis, we searched the NCBI database for *fem-1* gene sequences of other insects to conduct a multi-sequence alignment (Appendix A) and phylogenetic tree analysis (Appendix A). The results showed that the *Dafem-1* genes had the highest homology with the *Dendroctonus ponderosae fem-1* gene.

### 2.2. Analysis of Expression Levels of Dafem-1A, Dafem-1B, and Dafem-1C across Various Developmental Stages

The *Dafem-1*A, *Dafem-1*B, and *Dafem-1*C genes are expressed at different developmental stages of *D. armandi*, with varying expression patterns. No significant difference was found in the expression levels of the three genes during the larvae and pupae stages. However, in the adult stage, the expression levels of the *Dafem-1*A and *Dafem-1*C genes were significantly higher in females than in males, while the *Dafem-1*B gene was significantly higher in males than in females (Figure 1).

### 2.3. Analysis of Dafem-1A, Dafem-1B, and Dafem-1C Expression under Different Temperatures

In the larvae stage, the expression of the *Dafem-1*A gene was observed to be at a low level at both −10 °C and 0 °C, with no significant difference between the two. A significant increase in expression was noted at 10 °C, with a peak at 20 °C, followed by a significant decrease at 30 °C (Figure 2A). The *Dafem-1*B gene exhibited minimal expression at temperatures of −10 °C, 0 °C, and 30 °C, with no discernible difference between them. However, at 10 °C, there was a notable elevation in expression, reaching its peak at 20 °C (Figure 2B). The expression of the *Dafem-1*C gene was observed to be at a low level at both −10 °C and 0 °C, with no significant difference between the two temperatures. A significant increase in expression was noted at 10 °C, with a peak at 20 °C, followed by a significant decrease at 30 °C (Figure 2C).

In the pupae stage, the expression of the *Dafem-1*A gene was observed to be at a relatively low level at both 10 °C and 20 °C, with no significant difference between the two temperatures. However, a significant increase in expression was noted at 0 °C, 10 °C, and 30 °C (Figure 2D). The *Dafem-1*B gene exhibited a peak in expression at −10 °C, followed by a notable decline at 0 °C, 10 °C, 20 °C, and 30 °C, with a subsequent trough in expression at 30 °C (Figure 2E). The *Dafem-1*C gene expression exhibited a nadir at −10 °C, followed by a marked elevation at 0 °C, a decline at 10 °C, a further increase at 20 °C, and a subsequent decline at 30 °C (Figure 2F).

In the adult stage, male *Dafem-1*A gene expression was found to be significantly lower at −10 °C than at 0 °C, with a peak observed at 0 °C. There was a notable decline at 10 °C, 20 °C, and 30 °C, and a subsequent trough was reached at 30 °C. The expression of the female *Dafem-1*A gene was found to be significantly lower at 10 °C than that observed at 0 °C, reached a peak at 0 °C, and subsequently decreased significantly at 10 °C. Significant decreases in expression were observed at 20 °C and 30 °C, with no significant differences between the two temperatures. Significant differences in *Dafem-1*A gene expression between males and females were noted at −10 °C, 0 °C, 10 °C, and 20 °C, with no significant differences between the sexes at 30 °C (Figure 2G). It was observed that the expression of the *Dafem-1*B gene in males reached its lowest point at −10 °C, exhibiting a significant increase at 0 °C, a significant decrease at 10 °C, a significant increase at 20 °C, and a significant decrease once more at 30 °C. The expression level of the female *Dafem-1*B gene was observed to be lower at both −10 °C and 20 °C, with no significant difference in gene expression between the two temperatures. A significant increase in expression was observed at 0 °C, 10 °C, and 30 °C, with the expression level reaching its peak at 10 °C. There was a significant difference in the expression of the *Dafem-1*B gene between males and females under all temperature treatments (Figure 2H). The expression of the male *Dafem-1*C gene was not significantly different at either −10 °C or 0 °C, and both temperatures were below 10 °C, with the highest level of expression occurring at 10 °C. No significant difference was observed in expression at 20 °C and 30 °C, with both temperatures exhibiting significantly lower expression than 10 °C. The expression of the female *Dafem-1*C gene was found to be significantly lower at −10 °C than at 0 °C. No significant difference was observed in gene expression at 0 °C and 10 °C. Gene expression reached a trough at 20 °C, and expression was significantly higher at 30 °C. Significant differences in *Dafem-1*C gene expression levels were observed between males and females across all temperature treatments.

### 2.4. Analysis of Dafem-1A, Dafem-1B, and Dafem-1C Expression in Different Nutrients

To examine the impact of absent nutrients, such as protein, cellulose, and reducing sugar, on the expression levels of the *Dafem-1*A, *Dafem-1*B, and *Dafem-1*C genes, this study established four formulas to treat the *D. armandi* (the pupae do not eat). In larvae, the expression levels of the *Dafem-1*A, *Dafem-1*B, and *Dafem-1*C genes were significantly downregulated in comparison with the control group (FA) following treatment with a formula lacking protein (FB), cellulose (FC), and reducing sugar (FD). Furthermore, the greatest degree of downregulation was observed in the absence of protein formula (FB) treatment for both the *Dafem-1*A and *Dafem-1*B genes (Figure 3A–C). In adults, the expression level of the male *Dafem-1*A gene was significantly reduced following treatment with a formula lacking protein (FB), cellulose (FC), and reducing sugar (FD). The greatest downregulation was observed in the treatment group using the formula lacking protein (FB). The expression level of the female *Dafem-1*A gene was significantly reduced following the administration of a formula lacking protein (FB) and reducing sugar (FD), whereas no significant difference was observed in the expression level under formula treatment lacking cellulose (FC) (Figure 3D). The expression level of the male *Dafem-1*B gene was significantly reduced following treatment with a formula lacking protein (FB), cellulose (FC), and reducing sugar (FD), with the greatest downregulation observed in the formula lacking protein (FB). The expression level of the female *Dafem-1*B gene was significantly reduced following treatment with a formula lacking protein, while no significant difference in expression level was observed under the formula lacking cellulose (FC) and reducing sugar (FD) (Figure 3E). The expression level of the male *Dafem-1*C gene was significantly reduced following formula treatment lacking protein (FB), cellulose (FC), and reducing sugar (FD). The expression level of the female *Dafem-1*C gene was significantly reduced following treatment with formulas lacking protein (FB), cellulose (FC), and reducing sugar (FD), with the greatest downregulation observed under the formula treatment lacking protein (FB) (Figure 3F). Significant differences in the expression levels of the *Dafem-1*A, *Dafem-1*B, and *Dafem-1*C genes were evident between males and females in all formulas (FA, FB, FC, and FD). The expression level of the female *Dafem-1*A gene was higher than that of the male, the expression level of the male *Dafem-1*B gene was higher than that of the female, and the expression level of the female *Dafem-1*C gene was higher than that of the male.

### 2.5. Analysis of Dafem-1A, Dafem-1B, and Dafem-1C Expression in Feeding Duration Treatment

To investigate the effect of feeding duration on the expression of the *Dafem-1*A, *Dafem-1*B, and *Dafem-1*C genes, in this study, the starved larvae and adults of *D. armandi* were placed in a culture dish containing formula A, and the insects were extracted every 8 h to analyze their gene expression levels. In order to eliminate the influence of factors such as time, aging, age, and non-feeding on the duration of the feeding treatment, this study employed a negative control, which measured the expression levels of the three *Dafem-1* genes in *D. armandi* at 0 h, 8 h, 16 h, 24 h, 32 h, 40 h, and 48 h without feeding. The negative control treatment demonstrated that as time elapsed, there was no notable discrepancy in the expression levels of the three *Dafem-1* genes between larvae and adults (Appendix A). The results of the positive treatment indicated that the expression levels of the *Dafem-1*A, *Dafem-1*B, and *Dafem-1*C genes were significantly upregulated over time in the larvae stages in comparison to the control group, reaching their maximum values at 48 h (Figure 4A–C). The final outcome for adults was identical. In addition, during the adult stage, there were significant differences in the expression levels of the *Dafem-1*A, *Dafem-1*B, and *Dafem-1*C genes in males and females. The expression levels of the *Dafem-1*A and *Dafem-1*C genes were significantly higher in females than in males, while the expression levels of the *Dafem-1*B gene were significantly lower in females than in males (Figure 4D–F). The negative control treatment also produced the same experimental outcomes (Appendix A).

### 2.6. Analysis of Dafem-1A, Dafem-1B, and Dafem-1C Expression in Terpenoid Treatments

In order to investigate the effects of terpenoid treatments on *Dafem-1*A, *Dafem-1*B, and *Dafem-1*C gene expression, *D. armandi* was fumigated with seven terpenoids in the course of this study. In the larvae stage, the expression of the *Dafem-1*A gene was significantly reduced following the administration of terpenoids, with the exception of no significant difference observed in the case of (+)-α-pinene treatment (Figure 5A). In comparison to the control group, the expression levels of the *Dafem-1*B and *Dafem-1*C genes were significantly reduced following the administration of the seven terpenes (Figure 5B,C). Furthermore, the greatest degree of downregulation was observed in the *Dafem-1*C gene following (+)-camphene treatment (Figure 5C).

At the pupae stage, the expression of the *Dafem-1*A gene was significantly reduced in comparison to the control group following the administration of terpenoids, with the exception of no notable difference in *Dafem-1*A gene expression following (+)-α-pinene treatment (Figure 5D). The expression levels of the *Dafem-1*B and *Dafem-1*C genes were significantly reduced following the treatment of seven terpenoids. Among these, the greatest reduction was observed in the expression of the *Dafem-1*B gene following treatment with (+)-3-carene (Figure 5E,F).

In the adult stage, both the male and female *Dafem-1*A gene expression levels were significantly downregulated after treatment with seven terpenoids, when compared to the control group. Furthermore, there were significant differences in the male and female *Dafem-1*A gene expression levels between the groups (Figure 5G). In comparison to the control group, with the exception of no significant difference in the male *Dafem-1*B gene expression following treatment with (+)-α-pinene, all other terpenoids demonstrated a significant downregulation in the male *Dafem-1*B gene expression. In comparison to the control group, the expression level of the female *Dafem-1*B gene was found to be significantly downregulated following the administration of seven terpenoids. Additionally, notable discrepancies were observed in the expression levels of the male and female *Dafem-1*B genes between the groups (Figure 5H). In comparison to the control group, the expression levels of the male and female *Dafem-1*C genes were found to be significantly downregulated following treatment with seven terpenoids. Furthermore, notable discrepancies were observed in the expression levels of the *Dafem-1*C genes between the groups (Figure 5I).

### 2.7. Analysis of Dafem-1A, Dafem-1B, and Dafem-1C Expression in RNAi Treatment

To investigate the gene functions of *Dafem-1*A, *Dafem-1*B, and *Dafem-1*C, we performed RNA interference on these three genes at three developmental stages of *D. armandi*. The results demonstrated that with the increase in experimental days, the expression levels of the *Dafem-1*A, *Dafem-1*B, and *Dafem-1*C genes decreased gradually in the larvae, pupae, and adult stages (including females and males). On the third day, the expression levels of all three genes decreased to extremely low levels (Figure 6A–C). In the larvae, pupae, and adult of *D. armandi* (including males and females), after silencing the *Dafem-1*A gene, the expression of the *Dafem-1*C gene was significantly downregulated, while there was no significant difference in the expression level of the *Dafem-1*B gene (Figure 7A,B). After silencing the *Dafem-1*B gene, there was no significant difference in the expression levels of the *Dafem-1*A and *Dafem-1*C genes (Figure 7C,D). After silencing the *Dafem-1*C gene, the expression level of the *Dafem-1*A gene was significantly downregulated, while there was no significant difference in the expression level of the *Dafem-1*B gene (Figure 7E,F). After RNAi treatment during the larvae stage of *D. armandi*, the mortality and malformation rates increased significantly compared to the control group (Figure 8A,B). After interference with the *Dafem-1*A, *Dafem-1*B, and *Dafem-1*C genes, the emergence rate decreased significantly (Table 3). The interference of the three *Dafem-1* genes can lead to a sex imbalance, interference with the *Dafem-1*A and *Dafem-1*C genes can lead to an excess of males compared to females, while interference with the *Dafem-1*B gene can lead to an excess of females compared to males (Table 3).

## 3. Discussion

The *fem-1* gene is one of the most pivotal sex-determining genes in the nematode *C. elegans*. Spence et al. conducted a transcriptome sequencing of *C. elegans*, which revealed the function of the *fem-1* gene in *C. elegans* [8]. Furthermore, the completion of genome sequencing has resulted in the publication of the complete sequences of *fem-1* genes from a number of species, including *D. melanogaster*, *D. ponderosae*, *Bombyx mori*, *Tribolium castaneum*, and *Apis mellifera*, in the NCBI database. In this study, three *fem-1* genes were identified and cloned in *D. armandi*. Through amino acid multiple sequence alignment and phylogenetic tree analysis, the results showed that all three *fem-1* genes in *D. armandi* had the highest homology with the *D. ponderosae fem-1* gene, indicating a close genetic relationship between them.

The expression levels of the *Dafem-1* genes at different developmental stages showed that the *Dafem-1*A, *Dafem-1*B, and *Dafem-1*C genes were all expressed in the larvae, pupae, and adults of *D. armandi*. This suggests that the *Dafem-1* gene may be related to the growth and development of *D. armandi*. There was no significant difference in the expression levels of the three *Dafem-1* genes during the larvae and pupae stages. However, during the adult stage, the expression levels of the female *Dafem-1*A and *Dafem-1*C genes were higher than those of the male, while the *Dafem-1*B gene exhibited the opposite pattern. The outcome of this study is analogous to the findings of the research conducted by *C. elegans* on the *fem-1* gene, which is extensively expressed throughout the developmental stages of *C. elegans*.

The impact of temperature on the growth, development, and even sex determination of living organisms is significant. Prior research on *C. elegans* has demonstrated that specific alleles of the *fem-1* gene are temperature-sensitive [46,47,48]. The male-to-female sex ratio of adult *Sciara ocellaris* (a mosquito species) is approximately 50% when the environmental temperature during the egg stage is 18 °C and 20 °C. However, if the environmental temperature is 24 °C and 28 °C, the proportion of males decreases to 30–37% [49]. In the study of green turtles, a significant correlation was identified between temperature and the sex ratio. When the temperature is below 28.8 °C, there is a notable prevalence of males, whereas when the temperature is above 28.8 °C, there is a greater representation of females [50,51]. A comparable example can be found in the findings of Dai et al., which indicate a notable disparity in the sex ratio of adult *D. armandi* during the overwintering and non-overwintering periods. During the overwintering period, the number of females is considerably higher than that of males. During the winter season, there is a notable increase in the weight of both males and females. In comparison to the non-overwintering period, there are notable discrepancies in energy storage, detoxifying enzymes, and other variables [45]. However, the underlying cause of this phenomenon is not merely reflected in the level of biological activity, but also in the expression of genes, such as *Dafem-1*. Following temperature treatment, significant differences were observed in the gene expression levels of the *Dafem-1*A, *Dafem-1*B, and *Dafem-1*C genes, indicating that the *Dafem-1* genes exhibit differential responsiveness to varying temperatures. It can be observed that temperature has an impact on the physiological activity of *D. armandi*, as well as on the expression of the *Dafem-1* gene. Nevertheless, it remains unclear whether the observed effect of temperature on the sex ratio of *D. armandi* is mediated by the *Dafem-1* gene. Further investigation into the function of the *Dafem-1* genes may provide additional insights into this phenomenon. It can be concluded that temperature has a significant impact on the *Dafem-1* genes. Nutrients are essential for the life activities, growth, and development of insects. In this experiment, the *D. armandi* was treated with different nutrients and feeding duration. The results of different nutritional treatments demonstrated that compared with the control group, the expression levels of the *Dafem-1*A, *Dafem-1*B, and *Dafem-1*C genes were significantly downregulated when larvae, pupae, and adults were treated with feed lacking protein, cellulose, and reducing sugar. The largest downregulation was observed after treatment with feed lacking protein. Furthermore, in the context of the feeding time treatment, the *Dafem-1* genes demonstrated no significant difference between the control group (0 h) and the larvae, pupae, and adults (including males and females) at 8 h, with the increase in feeding time. From 16 h to 48 h, the gene expression levels exhibited a significant increase, reaching a peak at 48 h. The experimental group was administered formula A as a dietary supplement. The expression levels of the three *Dafem-1* genes exhibited an overall upward trend with increasing experimental time, and the differences were statistically significant. A similar outcome has been observed in research on nutrition in *Bactrocera tryoni*, *Anastrepha ludens*, and *Anastrepha oblique* [52,53,54]. This suggests that the consumption of nutrients is beneficial for the maintenance of active life activities, resulting in alterations to gene expression levels. Previous studies have demonstrated that starvation treatment of *D. armandi* has a significant impact on the downregulation of related genes, including juvenile hormone synthesis, mevalonate pathway, and neuropeptides [55,56,57]. Additionally, it affects the energy substrate content and flight ability of the organism [3]. Therefore, it can be concluded that the impact of nutrition on *D. armandi* is extremely broad. Consequently, the three *Dafem-1* genes of *D. armandi* exhibited notable differences in gene expression in response to the treatment of nutrients and feeding duration. These observed changes may be indirectly influenced by other biochemical and molecular pathways.

Following the administration of terpenoids, the expression levels of the *Dafem-1* genes were found to be significantly reduced in larvae, pupae, and adults of *D. armandi* in comparison to the control group. The terpenoids present in the body of *P. armandii* exert toxic effects on the *D. armandi*, including (+)-α-pinene, (−)-α-pinene, (+)-β-pinene, (−)-β-pinene, (+)-camphene, (+)-3-carene, and (±)-limone [58,59,60]. During the process of colonization of *P. armandii* by *D. armandi*, the toxicity of terpenoids in the host causes the downregulation of gene expression related to life activities, decreased metabolism, and even death [61,62]. Consequently, the impact of terpenoids on the *Dafem-1* genes of *D. armandi* may be attributed to their reduced life activity, which results in toxic effects and indirectly affects the expression level of *Dafem-1*, leading to a significant downregulation.

In addition, there were notable discrepancies in the gene expression profiles of the three *Dafem-1* genes between males and females following treatment with temperature, nutrition, feeding, and terpenoids. In consequence, the expression patterns remained uniform across all experimental conditions. The expression levels of the *Dafem-1*A and *Dafem-1*C genes were significantly higher in females than in males, whereas the expression level of the *Dafem-1*B gene was significantly higher in males than in females. This phenomenon may be related to the expression patterns of the three *Dafem-1* genes themselves.

In order to further investigate the function of the *Dafem-1* genes, we employed the RNAi interference method to silence the three *Dafem-1* genes. It is noteworthy that silencing of the *Dafem-1*A gene results in a downregulation of the *Dafem-1*C gene, while the silencing of the *Dafem-1*C gene leads to a downregulation of the *Dafem-1*A gene. The expression level of the *Dafem-1*B gene is not affected by the silencing of the *Dafem-1*A and *Dafem-1*C genes, and its own silencing does not affect the expression levels of the *Dafem-1*A and *Dafem-1*C genes. The observed outcome indicates the potential for a complex regulatory relationship among the three *Dafem-1* genes. To exclude the possibility that the *Dafem-1*A and *Dafem-1*C genes were knocked down simultaneously, we performed a concordance analysis of the dsRNA nucleotide sequences of the three *Dafem-1* genes, which showed that the concordance of the dsRNA nucleotide sequences of the *Dafem-1*A and *Dafem-1*C genes was only 39.96% (Appendix A) and that the nucleotides of the primer using the sequences were not concordant (Appendix A), so the possibility that the *Dafem-1*A and *Dafem-1*C genes were simultaneously interfered with was ruled out. However, the specific regulatory relationship between the *Dafem-1*A and *Dafem-1*C genes in this study was not found in any other species of *fem-1* genes, but similar experimental results were found in *D. melanogaster*. The numerous gene regulatory networks involved in *D. melanogaster* development include many mutually inhibitory interactions between two of these genes, and the mechanism of action is complex [63]. There is no evidence in this study that the *Dafem-1*A gene directly regulates the *Dafem-1*C gene, or that the *Dafem-1*C gene directly regulates the *Dafem-1*A gene, and further studies are needed to demonstrate the involvement of other genes. Moreover, following RNAi treatment, the mortality and deformity rates of *D. armandi* exhibited a notable increase, accompanied by a pronounced imbalance in the sex ratio. It is noteworthy that, following silencing, the number of males in the *Dafem-1*A and *Dafem-1*C groups declined significantly compared to the number of females, whereas in the *Dafem-1*B group, the opposite was observed. This result demonstrates a pronounced discrepancy compared to the control group. This evidence indicates that *Dafem-1* genes play a significant role in the growth and development of *D. armandi*, as well as influencing the development of both male and female individuals. Analogous experimental outcomes were observed in *E. sinensis* and *M. rosenbergii* [14,17]. The expression levels of the *Dafem-1*A and *Dafem-1*C genes are higher in females than males, whereas the *Dafem-1*B gene is higher in males than females. When the *Dafem-1*B gene is disrupted, the ratio of females to males is increased. Following the interference with the *Dafem-1*A and *Dafem-1*C genes, there was a greater prevalence of males than females. This result may be due to interference with the *Dafem-1* gene, which leads to changes in the development of male and female embryos, with female embryos developing into males and male embryos developing into females. RNA interference has been demonstrated to induce alterations in insect sex, as evidenced by the interference of *dsx* genes, which has been observed to result in changes in the sex of insects, including *Nilaparvata lugens* and *Plutella xylostella* [64,65]. However, there is currently no evidence that the *Dafem-1* genes are directly involved in male and female development in *D. armandi*. As previously stated, there is also an imbalance in the sex ratio between the overwintering and non-overwintering periods. During the overwintering period, there is a notable increase in the number of females. The environmental temperature during the overwintering period of *D. armandi* ranges from −10 °C to 10 °C [66,67]. It was observed that the expression levels of the female *Dafem-1*A and *Dafem-1*C genes were higher at 0 °C and 10 °C than at −10 °C, 20 °C, and 30 °C under different temperature treatments. It can therefore be concluded that there is a similarity between the data on expression levels and the phenotype. Nevertheless, there is currently no direct evidence to substantiate a positive correlation between temperature, the *Dafem-1* genes, and the sex ratio. Nevertheless, the phenomenon of the influence of temperature and *Dafem-1* gene expression on sex determination in *D. armandi* has prompted a more profound investigation into the underlying mechanisms of sex determination, including the interactions between intracellular proteins. The experiment validated the partial function of the three *Dafem-1* genes through RNAi but did not elucidate the manner by which the three *Dafem-1* genes interact with other genes and exert their function at the cellular level. In light of the interim results obtained from this experiment, it is evident that further evidence is required to elucidate the underlying mechanisms, which will also provide direction for further in-depth research. It is worth mentioning that RNAi strategies are widely used to manage pest population sizes with the aim of protecting crops [68] and forests [69]. However, the application of RNAi in *D. armandi* is only at the level of basic research; although some results have been obtained, more in-depth research is needed to achieve population management. The results of the *Dafem-1* genes have given us more information. If sterile mutants can be obtained through RNAi and released into the wild, the mutants will mate with normal insects, resulting in the production of sterile or lethal offspring. This will lead to a sharp increase in mortality and reproductive rates of the offspring, which will in turn result in a gradual decline in population size. This approach could be an effective method for managing the population of *D. armandi*. This is the ultimate goal of this work. However, there are still a lot of scientific questions that need to be explored, such as those at the cellular and molecular levels, which are the key directions of our subsequent research.

In conclusion, this study provides important information on the expression characteristics of the *fem-1* gene and its potential role in sex determination in *D. armandi*. The *fem-1* gene exhibits distinct response patterns to various environmental factors, including temperature, nutrition, feeding, and terpenoids. Additionally, its expression levels vary significantly between males and females. The silencing of the *Dafem-1*A, *Dafem-1*B, and *Dafem-1*C genes can affect *D. armandi*. The regulatory relationship between them remains an interesting phenomenon. This is of great significance for further exploration of the reasons why the *Dafem-1* genes affect the sex ratio of *D. armandi* and its regulatory relationship with other sex-determining genes. Such research will reveal the cascade relationship of sex determination in *D. armandi* and provide a theoretical basis for its prevention and management.

## 4. Materials and Methods

### 4.1. Insect Sample

The sample was obtained from the Huoditang experimental farm (E: 108°24′~108°29′, N: 33°18′~33°28′) of the Qinling forest ecological positioning station in Ningdong Forestry Bureau, Shaanxi Province. *P. armandii* that had been severely damaged by *D. armandi* in the area were selected, and their invasive wood segments were removed. The segments were then placed in a basin filled with wet sand, and the top of the wood was sealed with wax. A mesh was used to cover the outside of the wood segments. The larvae and pupae of the beetle were collected from the phloem of damaged wood. The male and female adults of the beetle were captured from the net and stored at −80 °C after emergence and flight. Three life stages of *D. armandi* were collected: larvae, pupae, and adults. Three replicates were established for each treatment, with 5 larvae and pupae selected for each treatment and 5 females or males selected for each treatment for adults. The sex of the adults was identified by the seventh abdominal tergite and external genitalia [70].

### 4.2. Total RNA Isolation and cDNA Synthesis

Total RNA was extracted from larvae, pupae, and adults using the Total RNA Extractor (Sangon, Shanghai, China). The extracted RNA was then synthesized into cDNA using the Hifair^®^ III 1st Strand cDNA Synthesis Kit (+gDNA wiper) (Vazyme, Nanjing, China) and stored at −20 °C until use. The integrity of the sample was verified through analysis on 1% agarose gels, with quantification conducted using a NanoDrop 2000 spectrophotometer (Thermo Scientific, Pittsburgh, PA, USA). The purity was determined through the calculation of the A260/A280 ratio (μg/mL = A260 × dilution factor × 40). Three replicates were established for each treatment, with 5 larvae and pupae selected for each treatment and 5 females or males selected for each treatment for adults.

### 4.3. Amplification of Genes, Cloning, and Sequence Analyses of Dafem-1

The cDNA synthesized from the sample was employed as a template for the polymerase chain reaction (PCR) reaction. The specific primers (Appendix A) were designed based on the *Dafem-1* gene sequences of other insects from the National Center for Biotechnology Information (NCBI) database http://www.ncbi.nlm.nih.gov/ 12 March 2024). Polymerase chain reaction (PCR) amplifications were conducted using a C1000 thermocycler (Bio-Rad, Hercules, CA, USA). cDNA amplification was performed in a 20 μL reaction volume, comprising 1 μL of cDNA, 0.25 μM of each primer, 10 μL of EcoTaq PCR SuperMix (TransGen Biotech, Beijing, China), and ddH_2_O added to the 20 μL reaction volume. The reaction conditions were as follows: the reaction was initiated at 94 °C for 5 min, followed by 30 cycles at 94 °C for 30 s; then, the melting temperature (TM) of each pair of primers was reacted for 30 s, followed by 72 °C for 30 s, with a final extension step. The PCR products were visualized on 1% agarose gels stained with 1× DuRed and compared with a 2 K plus DNA marker (TransGen Biotech, Beijing, China) for 10 min at 72 °C for 30 s. Single-stranded 5′ and 3′ RACE-ready cDNA were synthesized from RNA using a SMARTer RACE cDNA Amplification Kit (Clontech Laboratories Inc., Mountain, CA, USA) in accordance with the manufacturer’s instructions. The primer design employed partial sequences, and the PCR was conducted in accordance with the instructions provided in the SMARTer™ RACE cDNA Amplification Kit (Clontech Laboratories Inc., Mountain, CA, USA). The amplicons were purified, cloned, and sequenced. The resulting sequences were manually edited using EditSeq from DNASTAR 6.0.3.99 Softwareto obtain inserts, which were then BLASTed against the NCBI database. The complete sequences were compared with those deposited in GenBank using a BLASTP search.

### 4.4. Sequence Analyses of Dafem-1

The ProtParam program was used to determine the molecular mass (kDa) and isoelectric point (IP) of the two sequences. To gain further insight into the *Dafem-1* gene sequence, multiple sequence comparisons were conducted using the DNAMAN software. In addition to the three *Dafem-1* genes, fem-1 genes from seven species were included in the analysis, including *D. ponderosae* (*Dpfem-1*), *Sitophilus oryzae* (*Sofem-1*), *Anoplophora glabripennis* (*Agfem-1*), *Aethina tumida* (*Atfem-1*), *Cylas formicarius* (*Cffem-1*), *Tribolium madens* (*Tmfem-1*), and *Diorhabda sublineata* (*Dsfem-1*). Phylogenetic trees were constructed using the maximum likelihood method, as implemented in MEGA 11.0 [71,72]. In addition to the three *Dafem-1* genes, *fem-1* genes from 17 other species of Coleoptera were included in this study, including *D. ponderosae*, *S. oryzae*, *A. glabripennis*, *Anthonomus grandis grandis*, *A. tumida*, *C. formicarius*, *T. madens*, *Diorhabda sublineata, Diabrotica virgifera virgifera*, *Agrilus planipennis*, *Leptinotarsa decemlineata*, *Schistocerca gregaria*, *Photinus pyralis*, *Coccinella septempunctata*, *Onthophagus taurus*, *Gryllus bimaculatus*, and *Cimex lectularius*. The bootstrap values resulting from 500 pseudo-replicates are displayed at the nodes. The phylogenetic tree displays the bootstrap values (expressed as a percentage) at each branch point, with values below 50% omitted.

### 4.5. Different Environment Treatments of D. armandi

The collected beetles were divided into three developmental stages for this study: larvae, pupae, and adults. The *D. armandi* were exposed to temperature treatment (−10 °C, 0 °C, 10 °C, 20 °C, and 30 °C) for 24 h in a dark environment by the methods of Wang et al. [64,65] and Fu et al. [73]. The feed formula was modified in accordance with the recommendations set forth by Wang [74] for the treatment of *D. armandi*, with the objective of nutrient treatments and feeding duration treatments.

Substances added to all formulas (common components): 30 g of *P. armandi* phloem powder, 4 g of agar, 1 g of cholesterol, 0.2 g of inositol, 0.2 g of sodium chloride, 0.5 g of vitamin, 0.2 g of L(+)-ascorbic acid, 0.2 g of methylparaben, 0.2 g of potassium sorbate, and 20 g of distilled water.

Formula A (control): Add 10 g of peptone (protein), 10 g of cellulose, and 10 g of sucrose (reducing sugar);

Formula B: Add 10 g of cellulose and 10 g of sucrose (reducing sugar);

Formula C: Add 10 g of peptone (protein) and 10 g of sucrose (reducing sugar);

Formula D: Add 10 g of peptone (protein) and 10 g of cellulose.

The control group for the nutritional treatment was designated as formula A, while the treatment groups were identified as formulas B, C, and D. The lighting conditions were maintained at a low intensity, and the temperature was set at 10 °C. The objective was to investigate the effects of nutrients, including protein, cellulose, and reducing sugar, on the expression levels of the *Dafem-1* genes. For this purpose, first instar larvae that had just hatched from eggs and adult insects that had just emerged from pupae were selected and cultured in feed. Prior to RNA extraction, both larvae and adults were subjected to a dissection process to ascertain whether they engaged in feeding behavior.

To investigate the effects of protein, cellulose, and reducing sugar intake on *Dafem-1* gene expression, we selected first instar larvae that had just hatched from eggs and adult insects that had just emerged from pupae. These insects were then cultured in a dark environment at 10 °C for 0, 8, 16, 24, 32, 40, and 48 h in feed containing formula A. No feeding treatment was used as a negative control, and the treatment time was also 0, 8, 16, 24, 32, 40, and 48 h. The control groups for both positive and negative treatments had 0 h. The methods for treating terpenoids followed Dai et al. [62] and Liu et al. [2], using a concentration of 220 ppm for 24 h. The terpenoids used in this experiment were (+)-α-pinene, (−)-α-pinene, (+)-β-pinene, (−)-β-pinene, (+)-camphene, (+)-3-carene, and (±)-limone.

All treatments were conducted with three biological replicates. Each replicate consisted of 5 larvae or pupae, and 5 adult males and females. RNA was extracted immediately after treatment.

### 4.6. RNAi

The MEGA script T7 Transcription Kit was used to synthesize dsRNA based on the full length of the *Dafem-1* gene sequence (Appendix A). The final dsRNA product was diluted to 1000 ng/μL in DEPC-treated water and stored at −80 °C. The length of the *Dafem-1*A dsRNA product is 664bp, the length of the *Dafem-1*B dsRNA product is 571bp, and the length of the *Dafem-1*C dsRNA product is 494bp. The positions of the dsRNA primers for *Dafem-1*A, *Dafem-1*B, and *Dafem-1*C are illustrated in Appendix A. To synthesize dsRNA, 0.2 μL of the diluted product was injected into the abdomen of *D. armandi* using a 10 μL Hamilton microliter syringe (700 series, RN) and 32 g needle (Hamilton, Bonaduz, Switzerland). Each larvae or pupae was injected with 0.1 μL of dsRNA, while each adult was injected with 0.2 μL of dsRNA [75,76]. As a negative control, dsRNA of GFP was used. Blank controls were created using untreated beetles. Three treated beetles were randomly selected at 24, 48, and 72 h, immediately frozen in liquid nitrogen, and stored at −80 °C. The expression levels of the *Dafem-1* genes were analyzed using quantitative real-time PCR.

Following the injection of dsRNA, we monitored the survival of *D. armandi* larvae on a daily basis. Subsequently, gene expression levels were analyzed using quantitative real-time PCR and the following data were recorded: mortality, emergence, deformity rate, and sex ratio. The deformity observed following RNAi interference included a reduction in body size compared to that of normal insects, as well as aberrant pigmentation, manifesting as black–brown or dark yellow. Additionally, there was a notable absence of development in certain organs, such as the inability to develop wings or a sunken abdomen.

### 4.7. Quantitative Real-Time PCR

Specific primers for quantitative real-time polymerase chain reaction (qRT-PCR) were designed using the software Primer Premier 5.0, based on the nucleotide sequences obtained (Appendix A). To ensure that only a single product corresponding to the target sequence was amplified, a melting curve analysis was conducted. Prior to use, all primer pairs were tested to ensure that amplification efficiency was close to 100%. The expressions of the *β-actin* genes were employed as an internal control. Quantitative real-time PCR was conducted in triplicate in accordance with the manufacturer’s instructions using TransStart Top Green qPCR SuperMix (TransGen Biotech, Beijing, China) on a CFX96™ Quantitative real-time qPCR Detection System (Bio-Rad, Hercules, CA, USA). The quantitative real-time PCR was conducted according to the following program: the reaction was initiated at 95 °C for 10 min, followed by 40 cycles of denaturation at 95 °C for 5 s, annealing at the optimal temperature for each primer pair (Appendix A) for 15 s, and extension at 72 °C for 20 s. The relative expression levels were analyzed using the 2^−ΔΔCt^ method.

### 4.8. Statistical Analysis

All statistical data from gene expression analyses were performed using SPSS Statistics 23.0 (IBM, Chicago, CA, USA). All environmental treatment data were analyzed by one-way analysis of variance (ANOVA) and Tukey’s post hoc test to determine differences [77]. The relative expression levels of genes between males and females were compared using independent sample *t*-tests [78]. In addition, Prism 8.0.2 (GraphPad Prism Software, San Diego, CA, USA) was used to plot graphs. The particular data analysis techniques employed in the context of environmental processing were as follows:

#### 4.8.1. Different Developmental Stages

The relative expression levels of the *Dafem-1*A, *Dafem-1*B, and *Dafem-1*C genes were analyzed using one-way analysis of variance (ANOVA) and Tukey’s post hoc test to determine the differences in expression levels of the three *Dafem-1* genes at three developmental stages: larvae, pupae, and adult males. The same was true for larvae, pupae, and adult females. At the adult stage, the expression levels in male and female insects were assessed by independent sample *t*-tests.

#### 4.8.2. Different Temperature Treatments

The relative expression levels of the *Dafem-1*A, *Dafem-1*B, and *Dafem-1*C genes were analyzed using one-way analysis of variance (ANOVA) and Tukey’s post hoc test in order to ascertain the differences in relative gene expression levels of the three *Dafem-1* genes at the same developmental stages but at different temperatures. At the adult stage, expression levels in male and female insects were evaluated using independent sample *t*-tests for the same temperature treatments.

#### 4.8.3. Different Nutrient Treatments

The relative expression levels of the *Dafem-1*A, *Dafem-1*B, and *Dafem-1*C genes were analyzed using one-way analysis of variance (ANOVA) and Tukey’s post hoc test to determine the relative gene expression differences of the three *Dafem-1* genes at the same developmental stage but under different nutrients. At the adult stage, expression levels in male and female insects were assessed using independent sample *t*-tests for the same nutrient treatments.

#### 4.8.4. Different Feeding Duration Treatments

The relative expression levels of the *Dafem-1*A, *Dafem-1*B, and *Dafem-1*C genes were analyzed using one-way analysis of variance (ANOVA) and Tukey’s post hoc test to determine the relative gene expression differences of the three *Dafem-1* genes at the same developmental stage but under different feeding durations. At the adult stage, expression levels in male and female insects were assessed using independent sample *t*-tests for the same feeding duration treatments.

#### 4.8.5. Different Terpenoid Treatments

The relative expression levels of the *Dafem-1*A, *Dafem-1*B, and *Dafem-1*C genes were analyzed using one-way analysis of variance (ANOVA) and Tukey’s post hoc test to determine the relative gene expression differences of the three *Dafem-1* genes at the same developmental stage but under different terpenoid treatment durations. At the adult stage, expression levels in male and female insects were assessed using independent sample *t*-tests for the same terpenoid treatments.

#### 4.8.6. RNAi Treatments

The data on RNAi interference efficiency were subjected to analysis using one-way analysis of variance (ANOVA) and Tukey’s post hoc test. The relative expression levels of the three *Dafem-1* genes at the same developmental stage exhibited variability on different experimental days following RNA interference, which was conducted to assess the efficiency of RNAi.

The relative expression levels of the remaining two *Dafem-1* genes following the interference of one of the *Dafem-1* genes were analyzed using an independent sample *t*-test. This test was employed to illustrate the difference in the relative expression of the target *Dafem-1* gene between the pre-interference (control) and post-interference periods, with the interference of one of the genes present. Furthermore, the test was used to reveal the interplay between the three *Dafem-1* genes.

The degree of difference in mortality, deformity, emergence rate, and sex ratio was evaluated using independent sample *t*-tests. This data testing method was designed to explore the extent of differences in mortality, deformity rate, emergence rate, and sex ratio of *D. armandi* following targeted *Dafem-1* gene disruption compared to controls at the same developmental stage.

## Figures and Tables

**Figure 1 ijms-25-10349-f001:**
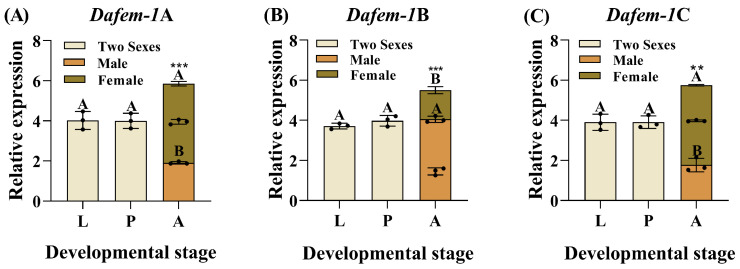
Relative expression of the *Dafem-1* genes in different developmental stages of *D. armandi*. (**A**) *Dafem-1*A; (**B**) *Dafem-1*B; and (**C**) *Dafem-1*C. The abbreviations used in the figure are L for Larvae, P for Pupae, and A for Adults. Relative expression levels were normalized to *β-actin*. All values are presented as mean ± SE (n = 3). Statistical analysis using one-way ANOVA and Tukey’s post hoc test revealed a significant difference at the *p* < 0.05 level. Capital letters indicate inter-group differences. The asterisk indicates a significant difference between males and females (** *p* < 0.01, and *** *p* < 0.001, independent sample *t*-test). Note: It is not possible to distinguish between larvae and pupae in terms of gender based on their morphology. Consequently, they are referred to as belonging to two sexes.

**Figure 2 ijms-25-10349-f002:**
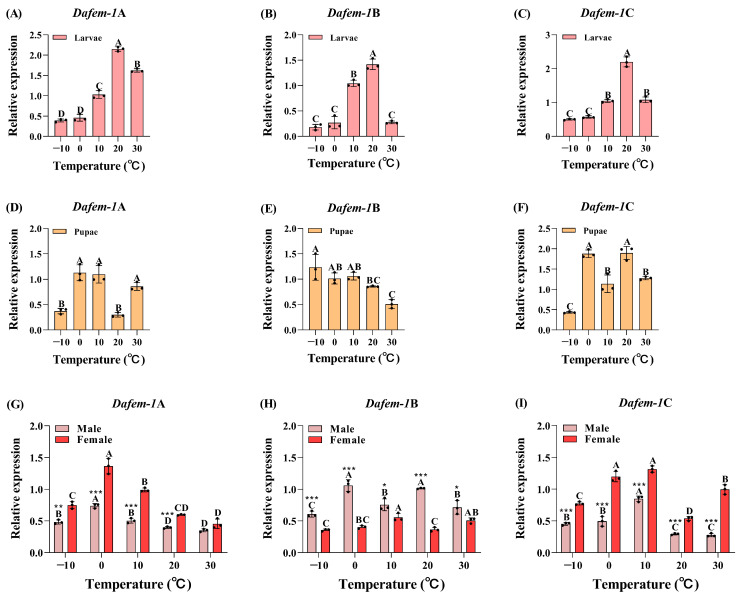
Relative expression of the *Dafem-1* genes in *D. armandi* under temperature treatments. (**A**) Larvae *Dafem-1*A; (**B**) larvae *Dafem-1*B; (**C**) larvae *Dafem-1*C; (**D**) pupae *Dafem-1*A; (**E**) pupae *Dafem-1*B; (**F**) pupae *Dafem-1*C; (**G**) adult *Dafem-1*A; (**H**) adult *Dafem-1*B; and (**I**) adult *Dafem-1*C. Relative expression levels were normalized to *β-actin*. All values are presented as mean ± SE (n = 3). Statistical analysis using one-way ANOVA and Tukey’s post hoc test revealed a significant difference at the *p* < 0.05 level. Capital letters indicate inter-group differences. The asterisk indicates a significant difference between males and females (* *p* < 0.05, ** *p* < 0.01, and *** *p* < 0.001, independent sample *t*-test).

**Figure 3 ijms-25-10349-f003:**
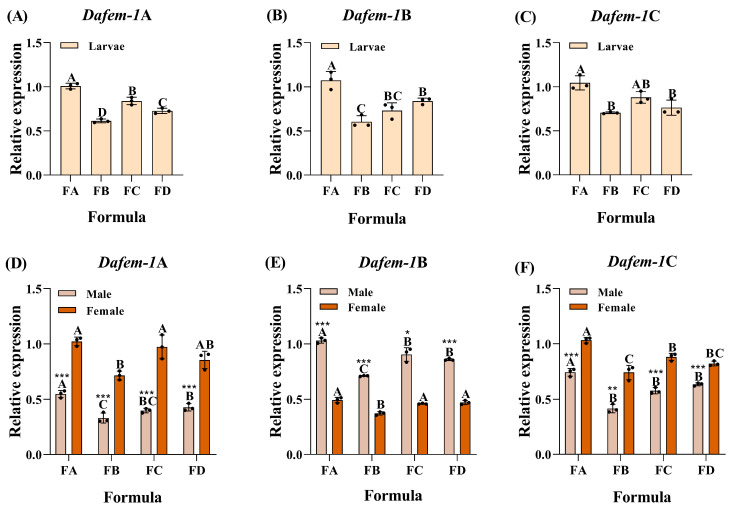
Relative expression of the *Dafem-1* genes in *D. armandi* under different nutrient treatments. (**A**) Larvae *Dafem-1*A; (**B**) larvae *Dafem-1*B; (**C**) larvae *Dafem-1*C; (**D**) adult *Dafem-1*A; (**E**) adult *Dafem-1*B; and (**F**) adult *Dafem-1*C. Formulas A, B, C, and D were designated as FA, FB, FC, and FD, respectively. Formula A served as a control, with additional protein, cellulose, and reducing sugar added to the feed in addition to the common components (see Section 4 for details). The treatment groups were formulas B, C, and D, which were based on formula A with protein, cellulose, and reducing sugar removed, respectively. Relative expression levels were normalized to *β-actin*. All values are presented as mean ± SE (n = 3). Statistical analysis using one-way ANOVA and Tukey’s post hoc test revealed a significant difference at the *p* < 0.05 level. Capital letters indicate inter-group differences. The asterisk indicates a significant difference between males and females (* *p* < 0.05, ** *p* < 0.01, and *** *p* < 0.001, independent sample *t*-test).

**Figure 4 ijms-25-10349-f004:**
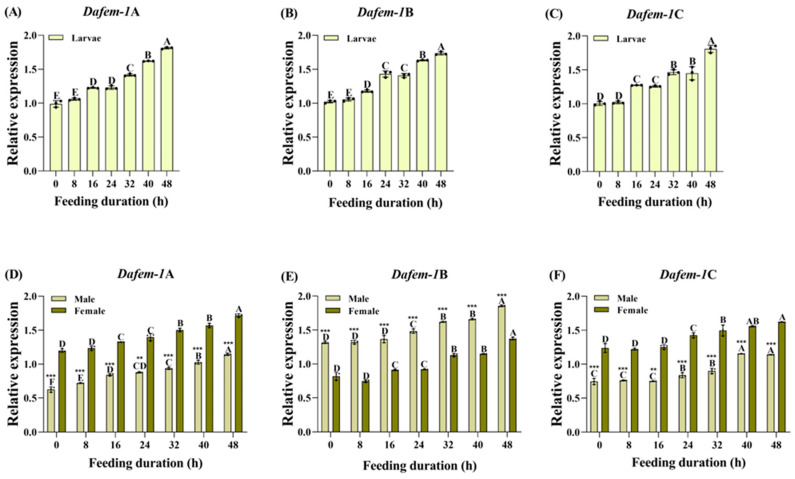
Relative expression of the *Dafem-1* genes in larvae and adults of *D. armandi* under feeding duration treatment. (**A**) Larvae *Dafem-1*A; (**B**) larvae *Dafem-1*B; (**C**) larvae *Dafem-1*C; (**D**) adult *Dafem-1*A; (**E**) adult *Dafem-1*B; and (**F**) adult *Dafem-1*C. Relative expression levels were normalized to *β-actin*. All values are presented as mean ± SE (n = 3). Statistical analysis using one-way ANOVA and Tukey’s post hoc test revealed a significant difference at the *p* < 0.05 level. Capital letters indicate the degree of difference between groups. The asterisk indicates a significant difference between males and females (** *p* < 0.01, and *** *p* < 0.001, independent sample *t*-test).

**Figure 5 ijms-25-10349-f005:**
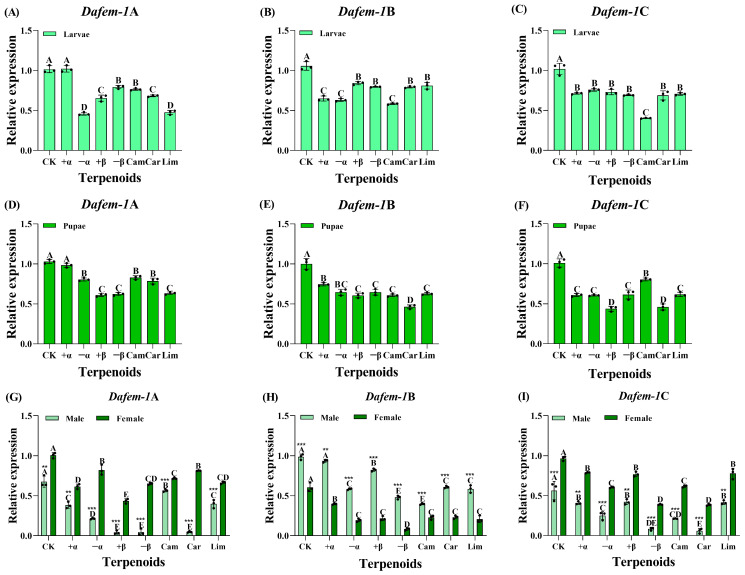
Relative expression of the *Dafem-1* genes in *D. armandi* under different terpenoid treatments. (**A**) Larvae *Dafem-1*A; (**B**) larvae *Dafem-1*B; (**C**) larvae *Dafem-1*C; (**D**) pupae *Dafem-1*A; (**E**) pupae *Dafem-1*B; (**F**) pupae *Dafem-1*C; (**G**) adult *Dafem-1*A; (**H**) adult *Dafem-1*B; and (**I**) adult *Dafem-1*C. CK for control check, +α for (+)-α-pinene, −α for (−)-α-pinene, +β for (+)-β-pinene, −β for (−)-β-pinene, Cam for (+)-camphene, Car for (+)-3-carene, and Lim for (±)-limone. The control check was not subjected to any terpenoid treatment. Relative expression levels were normalized to *β-actin*. All values are presented as mean ± SE (n = 3). Statistical analysis using one-way ANOVA and Tukey’s post hoc test revealed a significant difference at the *p* < 0.05 level. Capital letters indicate the degree of between-group differences. The asterisk indicates a significant difference between males and females (** *p* < 0.01, and *** *p* < 0.001, independent sample *t*-test).

**Figure 6 ijms-25-10349-f006:**
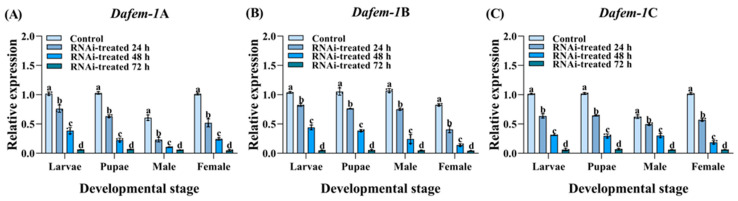
RNA interference efficiency detection of the *Dafem-1* genes in *D. armandi*. (**A**) *Dafem-1*A; (**B**) *Dafem-1*B; and (**C**) *Dafem-1*C. The control group was administered an equal volume of diethyl pyrocarbonate (DEPC) in water. Relative expression levels were normalized to *β-actin*. All values are presented as mean ± SE (n = 3). Statistical analysis using one-way ANOVA and Tukey’s post hoc test revealed a significant difference at the *p* < 0.05 level. Lowercase letters indicate the degree of within-group differences.

**Figure 7 ijms-25-10349-f007:**
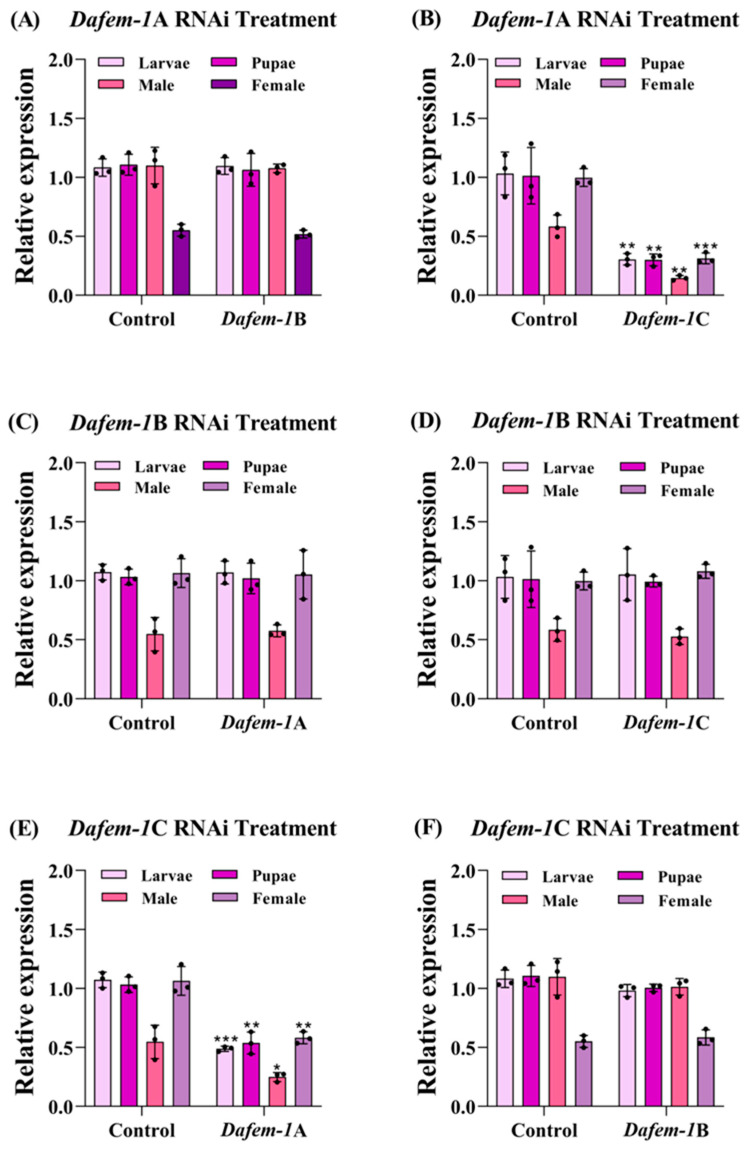
Relative expression of the three *Dafem-1* genes after 72 h of RNA interference. (**A**) Gene expression of *Dafem-1*B after *Dafem-1*A interference; (**B**) gene expression of *Dafem-1*C after *Dafem-1*A interference; (**C**) gene expression of *Dafem-1*A after *Dafem-1*B interference; (**D**) gene expression of *Dafem-1*C after *Dafem-1*B interference; (**E**) gene expression of *Dafem-1*A after *Dafem-1*C interference; and (**F**) gene expression of *Dafem-1*B after *Dafem-1*C interference. Relative expressions were normalized to *β-actin*. All values are presented as mean ± SE (n = 3). The asterisk indicates the significant difference between the RNAi interference experimental group and the control group (* *p* < 0.05, ** *p* < 0.01, and *** *p* < 0.001, independent sample *t*-test).

**Figure 8 ijms-25-10349-f008:**
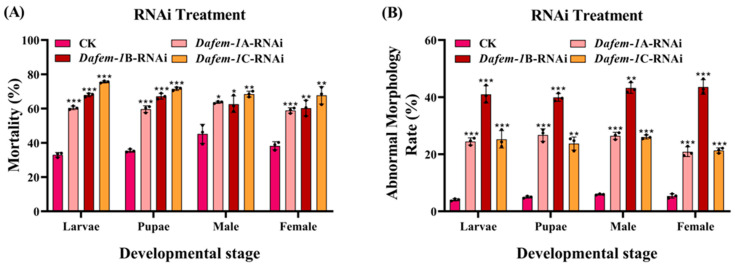
Mortality and deformity rates of *D. armandi* at different developmental stages after RNA interference of the three *Dafem-1* genes. CK for control check. (**A**) Mortality rates of larvae, pupae, and adults after RNA interference with the *Dafem-1*A, *Dafem-1*B, and *Dafem-1*C genes; (**B**) abnormal morphology rate of larvae, pupae, and adults after RNA interference with the *Dafem-1*A, *Dafem-1*B, and *Dafem-1*C genes. The control check was not subjected to RNAi treatment. All values are presented as mean ± SE (n = 3). The asterisk indicates the significant difference between the RNAi interference experimental group and the control group (* *p* < 0.05, ** *p* < 0.01, and *** *p* < 0.001, independent sample *t*-test).

**Table 1 ijms-25-10349-t001:** Physicochemical properties of the *Dafem-1*A, *Dafem-1*B, and *Dafem-1*C genes.

Gene Name	ORF Size (Aa/Bp) ^a^	Mw (kDa) ^a^	IP ^a^
*Dafem-1*A	630/2180	17.86	4.92
*Dafem-1*B	662/2307	18.92	4.90
*Dafem-1*C	650/2173	17.84	4.92

ORF, Open Reading Frame; MW, molecular weight; and IP, isoelectric point. ^a^ As predicted by the ProtParam program.

**Table 2 ijms-25-10349-t002:** Percent identities of the *Dafem-1*A, *Dafem-1*B, and *Dafem-1*C genes.

Gene Name	*Dafem-1*A	*Dafem-1*B	*Dafem-1*C
*Dafem-1*A	-	43.04% ^a^/38.13% ^b^	45.31% ^a^/31.87% ^b^
*Dafem-1*B	49.17% ^a^/38.13%^b^	-	43.04% ^a^/26.09% ^b^
*Dafem-1*C	45.31% ^a^/31.87%^b^	49.17% ^a^/26.09% ^b^	-

^a^ Percent identity of the *Dafem-1* nucleotide sequence. ^b^ Percent identity of the *Dafem-1* protein sequence.

**Table 3 ijms-25-10349-t003:** Emergence rates and sex ratios of *D. armandi*.

Groups	Emergence Rate (%)	Sex Ratio (Male:Female)
CK	60.9	3.2:2.89
*Dafem-1*A-RNAi	26.9	3.1:1.2
*Dafem-1*B-RNAi	24.6	0.9:3.7
*Dafem-1*C-RNAi	25.4	2.3:0.5

## Data Availability

The datasets generated and/or analyzed during the current study are available from the corresponding author upon reasonable request.

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
