# Peer review of "Fem-1 Gene of Chinese White Pine Beetle (Dendroctonus armandi): Function and Response to Environmental Treatments"

_ijms, 2024, doi:10.3390/ijms251910349_

Round 1
Reviewer 1 Report
Comments and Suggestions for Authors
The authors report on molecular characterization of three Fem1 genes in the white pine beetle, Dendroctonus armandi. Full length clones are produced for each of the three gene products and sequences are compared. With qPCR, gene expression differences are evaluated under several different environmental conditions. Finally, RNAi is used to knockdown expression of each of the three fem1 genes, and associated phenotypes are assessed. The manuscript has some interesting findings that are worthy of publication. However, in several aspects the reporting is substantially flawed, and extensive revisions are required before this manuscript would be suitable for publication.
Importantly, the most clear findings in this report are that Fem1 A and C are upregulated in females versus males, and Fem1-B is upregulated in males versus females… and that these observations correlate directly with sex-ratio distortions when each of these three genes are knocked down. That is, when A and C are knocked down, there is a clear bias towards males, and when B is knocked down, there is a clear bias towards females. However, these facts are not even mentioned in the abstract, and are given very little direct commentary in the Discussion. This is a glaring omission, that the strongest factual line of evidence for the role of these genes is not more prominently featured in this report.
In the Introduction, from lines 43 to 62, in the paragraph on Fem1, there is some mention that Fem1 is an intracellular protein, that is likely involved in protein-protein interactions and that there is some evidence in different species, that there are bisexual expression, with roles in spermatogenesis, ovarian function and sex determination. That is fine, however, there is little substance as it relates to the actual molecular/cellular function of the fem1 genes. This is important to highlight because in this study, RNAi is conducted, and phenotypes are observed, but there is no clear discussion on how gene knockdown could be causing the observed phenotypes at the cellular/molecular level. It the precise cellular/molecular functions of these genes are not known, it should be clearly stated.
Also In the introduction, there are instances of logically flawed statements.
For example, from lines 36 to 40, there is a gap in the logic from the first sentence to the second one. It is not clear how understanding mechanisms of sex determination relates to the phenomena of male attraction to female pheromone and how thus knowing the mechanism of sex determination could lead to population control.
On lines 57-59, it is mentioned that fem-1 was identified in two species, therefore suggesting that the fem-1 gene may be involved in ovarian maturation and development. There seems to be missing information here, related to what -beyond identification- suggests the genes may be involved in ovarian function.
In Figure 1, in each diagram, one of the conditions is “two sexs”. This should be “two sexes”. And furthermore, it is not clear what this means. Was it meant to say “mixed sexes”? Whatever was intended, it should also be clearly defined in the Figure Legend.
Lines 108-118 – the description of the results here is not so informative, to simply describe for each gene and condition what the temperature was for maximum and minimum expression levels were. It would be more informative to specify fold change differences across the temperature range.
Also, in Figure 2, across several panels, such as D and F, for example, but also in other cases, it is not correct to say that the gene expression reaches its maximum or minimum at a single temperature, when it is clear that there are multiple statistically indistinguishable temperatures where maximum and or minimum gene expression values are observed. For example, in panel D, statistically same maximums are observed at 0, 10 and 30, and statistically same minimums are observed at -10 and 20. It cannot so simply be written that the maximum is observed at 0 and the minimum is observed at 20, as is written on lines 115-116. The statistics do not support statements such as these.
Line 123. “male and male” – this appears to be a typo.
Line 142-143. “significantly downregulated in the treatment group compared to the control group.”
It is not clear here at this point what the different treatment groups are nor what the control group is. It is also not clear in Figure 3 what the different diets are. They are defined as A,B,C,D, but there is no mention in the figure legend what these diets represent. They are defined in the methods section at the end of the manuscript, but information presented in the figure should stand alone. The diets need to be described in the figure legend.
Line 147-148. “the gene expression of Fem1A/B/C was found to be significantly downregulated”
The context is missing here. Downregulated in what condition compared to what other condition?
Section 2.5. Analysis of Dfem-1A/B/C expression in feeding duration treatment
For this set of experiments, how can it be known that you are measuring effect of feeding duration and not just effect of age? It is not clear that there was a comparison made to non-feeding insects over the same time period. In the discussion there is some mention of effects of non-feeding, or starvation, on gene expression (lines 322-328), but it is not clear that this variable was considered in this set of experiments. In looking at Figure 4, the results that gene expression increases over time, but it is not clear if the cause can be isolated to effect of feeding versus effect of time, or aging.
Line 223-229. The results show that silencing A results in down-regulation of C, and silencing C results in down-regulation of A. As the report is written in structured now, these results presented, as such, begs the question of how similar A, B, and C are? Because it could be the case that A and C are so highly similar that targeting one for knockdown also knocks down the other. In looking at the supplemental data on the alignments and the phylogeny, there are indications that this should not be the case, but this is not clear at all in the main text of the mauscript. Instead of just mentioning S5 and S6 in Results Section 2.1, the actual data on how similar the different Fem1 genes are should be explicitly mentioned in terms of percentage identity, for example.
Related to this, In Table 2, it is reported that emergence rate is not affected compared to controls when A is knocked down, but it is when B and C are knocked down. But if knock down of A results in down-regulation of C, and vice versa, how is it accounted for that directly targeting C, but not A, can affect the emergence rate? This needs to be addressed in the discussion section, and this ties into a need for more text in the introduction on the cellular and molecular roles of these genes, if such information is known.
For Figure 7, it is not clear at what time point subsequent to RNAi injections the data is being shown for. How many hours/days after injections? This should be mentioned in the figure legend.
In Figure 8, and in associated text in the results and materials and methods, it is not clear how abnormal morphology is assessed. This needs to be more clearly defined in the appropriate Materials and Methods section.
In the Discussion:
Line 281. “This result is similar to that observed in the fem-1 gene in C. elegans?
Similar how? This should be clearly described.
Lines 293-294, and following discussion “indicates a significant imbalance in the sex-ratio between male and female adults during the over-wintering and non-overwintering periods.”
The connection between fem1 gene expression, sex ratio distortions and overwintering and non-overwintering populations need to be made clearer. Are the distorted sex ratios in these populations due to differences in fem1 gene expression, directly? Or do they even correlate? This is not clear. Unless all of these different factors are shown to be directly caused by or influenced by fem1 expression or fem1 protein function, the correlations can be spurious. Correlation is not the same as causation.
Line 317-318. “The experimental group was administered formula A as a dietary supplement.”
Up until now, Formula A has not been defined, so it is not clear what this is. Furthermore, in the methods section, there is an indication that Formula A is the control treatment, not an experimental one.
Lines 368-396. Most of the text here is inappropriate for the Discussion. If anything, there should be a proper introduction of RNAi theme in the Introduction section and the results of this study should be properly contextualized in the discussion section. As it is written now, nearly 30 lines of text are provided here, most of which amount to a literature review of RNAi in C. elegans. This is inappropriate for this article.
In the Materials and Methods:
Under Section 4.1 or 4.2, there needs to be an indication of how many individuals were collected for each sample type.
The section 4.8 on qRT-PCR is very incomplete. It mentions that qRT-PCR was carried out as previously described. But there is no reference to the previous description. Even if there was a reference here, there is no mention of thermocycling conditions, nor is there any mention of how relative gene expression values are calculated. This is not acceptable methods reporting for qRT-PCR data.
Author Response
Comments 1: Importantly, the most clear findings in this report are that Fem1A and C are upregulated in females versus males, and Fem1B is upregulated in males versus females and that these observations correlate directly with sex-ratio distortions when each of these three genes are knocked down. That is, when A and C are knocked down, there is a clear bias towards males, and when B is knocked down, there is a clear bias towards females. However, these facts are not even mentioned in the abstract, and are given very little direct commentary in the Discussion. This is a glaring omission, that the strongest factual line of evidence for the role of these genes is not more prominently featured in this report.
Response 1: Thank you for your comment. The mutual regulatory relationship between the three genes Dafem-1 has been included in the abstract, and a detailed analysis has been conducted in the Discussion section. The pertinent modifications are reflected in lines 24-28, 499-512.
Comments 2: In the Introduction, from lines 43 to 62, in the paragraph on Fem1, there is some mention that Fem1 is an intracellular protein, that is likely involved in protein-protein interactions and that there is some evidence in different species, that there are bisexual expression, with roles in spermatogenesis, ovarian function and sex determination. That is fine, however, there is little substance as it relates to the actual molecular/cellular function of the fem1 genes. This is important to highlight because in this study, RNAi is conducted, and phenotypes are observed, but there is no clear discussion on how gene knockdown could be causing the observed phenotypes at the cellular/molecular level. It the precise cellular/molecular functions of these genes are not known, it should be clearly stated.
Response 2: Thank you for your comment. Given the limitations of our research, which did not delve into the cellular and molecular levels, we have provided additional explanations in the Discussion regarding the cellular and molecular function of the Dafem-1 gene. The pertinent modifications are reflected in lines 537–545.
Comments 3: Also In the introduction, there are instances of logically flawed statements. For example, from lines 36 to 40, there is a gap in the logic from the first sentence to the second one. It is not clear how understanding mechanisms of sex determination relates to the phenomena of male attraction to female pheromone and how thus knowing the mechanism of sex determination could lead to population control.
Response 3: Thank you for your comment. The objective of this study is to present the distinctive physiological attributes associated with the invasion of D. armandi into its host organisms and to elucidate the relationship between this phenomenon and the experimental outcomes. This will facilitate the introduction of the concept that the aim of this experiment is to establish a theoretical foundation for the population management of D. armandi. The pertinent modifications are reflected in lines 42-47.
Comments 4: On lines 57-59, it is mentioned that fem-1 was identified in two species, therefore suggesting that the fem-1 gene may be involved in ovarian maturation and development. There seems to be missing information here, related to what -beyond identification- suggests the genes may be involved in ovarian function.
Response 4: Thank you for your comment. The aforementioned sentence has been presented in greater detail. The pertinent modifications are reflected in lines 65-74.
Comments 5: In Figure 1, in each diagram, one of the conditions is “two sexs”. This should be “two sexes”. And furthermore, it is not clear what this means. Was it meant to say “mixed sexes”? Whatever was intended, it should also be clearly defined in the Figure Legend.
Response 5: Thank you for your comment. Given the inability to definitively ascertain the gender of the D. armandi during the larvae and pupae stages, we have elected to categorise it as "Two sexes". This modification is reflected in Figure 1 and further elucidated in the Figure Legend. The pertinent modifications are reflected in lines 146-147.
Comments 6: Lines 108-118 – the description of the results here is not so informative, to simply describe for each gene and condition what the temperature was for maximum and minimum expression levels were. It would be more informative to specify fold change differences across the temperature range. Also, in Figure 2, across several panels, such as D and F, for example, but also in other cases, it is not correct to say that the gene expression reaches its maximum or minimum at a single temperature, when it is clear that there are multiple statistically indistinguishable temperatures where maximum and or minimum gene expression values are observed. For example, in panel D, statistically same maximums are observed at 0, 10 and 30, and statistically same minimums are observed at -10 and 20. It cannot so simply be written that the maximum is observed at 0 and the minimum is observed at 20, as is written on lines 115-116. The statistics do not support statements such as these.
Response 6: Thank you for your comment. The content presented in this section is not rigorous. In light of the aforementioned opinions, in addition to rectifying the issues identified in section 2.3 of the results, we also assessed the consistency between the descriptions in other results sections and the fundamental statistical analysis, implementing the necessary alterations. The pertinent modifications are reflected in lines 149-255.
Comments 7: Line 123. “male and male” – this appears to be a typo.
Response 7: Thank you for your comment. The aforementioned writing error has been rectified through the rewriting of sections 2.3 and 2.4.
Comments 8: Line 142-143. “significantly downregulated in the treatment group compared to the control group.” It is not clear here at this point what the different treatment groups are nor what the control group is. It is also not clear in Figure 3 what the different diets are. They are defined as A,B,C,D, but there is no mention in the figure legend what these diets represent. They are defined in the methods section at the end of the manuscript, but information presented in the figure should stand alone. The diets need to be described in the figure legend.
Response 8: Thank you for your comment. The objective of this experimental section is to investigate the impact of exogenous proteins, cellulose, and reducing sugars on the expression level of Dafem-1. Accordingly, four private chat formulas have been established, each of which is based on reference [73] and has practical applicability. Following the addition of the common ingredients, two of the three substances, namely peptone (protein), cellulose and sucrose (reducing sugar), were separately added in order to investigate the effect of the missing components on the expression of Dafem-1. Formula A serves as the control group, with the application of peptone (protein), cellulose, and sucrose (reducing sugar). Formulas B, C, and D represent the treatment groups. Formula B incorporates cellulose and sucrose (reducing sugar), while Formula C comprises peptone (protein) and sucrose (reducing sugar). Formula D contains peptone (protein) and cellulose. The volume has been supplemented in the Materials and Methods section. Furthermore, additional information has been supplied to Figure 3. The pertinent modifications are reflected in lines 672-692.
Comments 9: Line 147-148. “the gene expression of Fem1A/B/C was found to be significantly downregulated”. The context is missing here. Downregulated in what condition compared to what other condition?
Response 9: Thank you for your comment. It has been observed that the wording of this section is inadequate and therefore a decision has been taken to rewrite the results section of this chapter in order to more accurately present the findings.
Comments 10: Section 2.5. Analysis of Dafem-1A/B/C expression in feeding duration treatment. For this set of experiments, how can it be known that you are measuring effect of feeding duration and not just effect of age? It is not clear that there was a comparison made to non-feeding insects over the same time period. In the discussion there is some mention of effects of non-feeding, or starvation, on gene expression (lines 322-328), but it is not clear that this variable was considered in this set of experiments. In looking at Figure 4, the results that gene expression increases over time, but it is not clear if the cause can be isolated to effect of feeding versus effect of time, or aging.
Response 10: We are grateful for your contribution to this discussion. The initial objective of this experiment was to examine the impact of diverse nutrients on Dafem-1 gene expression, as detailed in section 2.4. It is important to note that there is a distinction between the various types of nutrient, although intake is also a factor that must be considered. Accordingly, a feeding duration was established for the purpose of investigating alterations in Dafem-1 gene expression. In designing the experiment, consideration was given to factors such as starvation and age, in order to ensure the most accurate results. D. armandi does not ingest food during the egg and pupal stages. Consequently, the eggs and pupae were initially cultivated separately to facilitate the hatching of larvae and adults, and to ensure that they were not fed prior to processing. The data for the control group were obtained when the insects were not provided with food. Negative controls were incorporated into this segment of the experiment to preclude the potential for interference from extraneous factors, thereby facilitating a more comprehensive understanding of this aspect of the study (for a detailed illustration, please refer to Figure S7). The growth of D. armandi is relatively slow at 10°C, and a full life cycle (developmental stage) cannot be completed within 48 hours. Moreover, references 65 and 66 have cited pertinent research on the temperature treatment of D. armandi. Moreover, the methodology employed in relation to the duration of feeding has been enhanced and refined with a view to facilitating a more comprehensive understanding of the subject matter. The relevant amendments are set out in lines 693–699.
Comments 11: Line 223-229. The results show that silencing A results in down-regulation of C, and silencing C results in down-regulation of A. As the report is written in structured now, these results presented, as such, begs the question of how similar A, B, and C are? Because it could be the case that A and C are so highly similar that targeting one for knockdown also knocks down the other. In looking at the supplemental data on the alignments and the phylogeny, there are indications that this should not be the case, but this is not clear at all in the main text of the manuscript. Instead of just mentioning S5 and S6 in Results Section 2.1, the actual data on how similar the different Fem1 genes are should be explicitly mentioned in terms of percentage identity, for example.
Response 11: Thank you for your comment. The comment has been subjected to careful consideration and a nucleotide/protein sequence consistency analysis has been conducted on the three Dafem-1 genes separately. The results are presented in Table 2, and Table S2. The aforementioned results, when considered alongside the positions of the dsRNA primers for the Dafem-1A and Dafem-1C genes as illustrated in Figures S1 and S3, as well as the sequences between the dsRNA primers, indicate that the Dafem-1A and Dafem-1C genes will not be simultaneously subjected to interference. The pertinent modifications are reflected in lines 113-121, 129-131, 499-512.
Comments 12: In Table 2, it is reported that emergence rate is not affected compared to controls when A is knocked down, but it is when B and C are knocked down. But if knock down of A results in down-regulation of C, and vice versa, how is it accounted for that directly targeting C, but not A, can affect the emergence rate? This needs to be addressed in the discussion section, and this ties into a need for more text in the introduction on the cellular and molecular roles of these genes, if such information is known.
Response 12: We are most grateful to you for taking the time to leave a comment. We extend our sincerest apologies for the error that was made in entering this data. A review of the raw data revealed that the mortality rate following interference with the Dafem-1A gene was 26.9%, rather than 56.9%. The original data indicates that the number of individuals who have emerged is 56, and the total number of samples taken is 208. We extend our sincerest apologies for any confusion or misunderstanding that may have arisen due to our own errors. We are grateful for your attention in identifying and correcting the errors in our experiment. Consequently, we have amended the data in Table 2 and made corresponding corrections in the Results and Discussion sections. The pertinent modifications are reflected in lines 345-348 and Table 3.
Comments 13: For Figure 7, it is not clear at what time point subsequent to RNAi injections the data is being shown for. How many hours/days after injections? This should be mentioned in the figure legend.
Response 13: Thank you for your comment. The requisite temporal data has been incorporated into the Figure Legend. The pertinent modifications are reflected in lines 360-361.
Comments 14: In Figure 8, and in associated text in the results and materials and methods, it is not clear how abnormal morphology is assessed. This needs to be more clearly defined in the appropriate Materials and Methods section.
Response 14: Thank you for your comment. The morphological abnormalities observed following RNAi interference include a reduction in body size compared to that of normal insects, as well as aberrant pigmentation, manifesting as black brown or dark yellow. Additionally, there is a notable absence of development in certain organs, such as the inability to develop wings or a sunken abdomen, which impairs the ability to crawl normally. The aforementioned information has been subjected to further refinement in the section dedicated to materials and methods. The pertinent modifications are reflected in lines 725-728.
Comments 15: In the Discussion: Line 281. “This result is similar to that observed in the fem-1 gene in C. elegans? Similar how? This should be clearly described.
Response 15: Thank you for your comment. A correction has been made to this statement, and it should be noted that the fem-1 gene is widely expressed in all developmental stages of C. elegans. The pertinent modifications are reflected in lines 394-396.
Comments 16: Lines 293-294, and following discussion “indicates a significant imbalance in the sex-ratio between male and female adults during the over-wintering and non-overwintering periods.” The connection between fem1 gene expression, sex ratio distortions and overwintering and non-overwintering populations need to be made clearer. Are the distorted sex ratios in these populations due to differences in fem1 gene expression, directly? Or do they even correlate? This is not clear. Unless all of these different factors are shown to be directly caused by or influenced by fem1 expression or fem1 protein function, the correlations can be spurious. Correlation is not the same as causation.
Response 16: Thank you for your comment. The phenomenon of sex ratio imbalance was discussed in greater detail during the overwintering period, based on the results of our RNA interference experiment. Furthermore, we explicitly delineated the constraints inherent to our research findings. The pertinent modifications are reflected in lines 398-423.
Comments 17: Line 317-318. “The experimental group was administered formula A as a dietary supplement”. Up until now, Formula A has not been defined, so it is not clear what this is. Furthermore, in the methods section, there is an indication that Formula A is the control treatment, not an experimental one.
Response 17: Thank you for your comment. After careful consideration, we have decided to delete this sentence, as it has little relevance to the context and may instead interfere with the reader's understanding. Meanwhile, we have provided a more detailed description in the Materials and methods section to clearly express our intention. As we answered in our previous comments, our experiment not only investigated the effects of different nutrients on Dafem-1 gene expression, but also aimed to investigate the effects of different intakes of the same nutrient on Dafem-1 gene expression. Therefore, we chose Formula A as the diet for the treatment of insects. The pertinent modifications are reflected in lines 686-692.
Comments 18: Lines 368-396. Most of the text here is inappropriate for the Discussion. If anything, there should be a proper introduction of RNAi theme in the Introduction section and the results of this study should be properly contextualized in the discussion section. As it is written now, nearly 30 lines of text are provided here, most of which amount to a literature review of RNAi in C. elegans. This is inappropriate for this article.
Response 18: Thank you for your comment. We have provided an appropriate description to RNAi in Introduction and, in addition, we have refined the RNAi in the Discussion section and provided an in-depth analysis related to our research. The pertinent modifications are reflected in lines 545-554.
Comments 19: In the Materials and Methods, Under Section 4.1 or 4.2, there needs to be an indication of how many individuals were collected for each sample type.
Response 19: Thank you for your comment. We have added this detail in section 4.1 and 4.2. The pertinent modifications are reflected in lines 611-613, 623-624.
Comments 20: The section 4.8 on qRT-PCR is very incomplete. It mentions that qRT-PCR was carried out as previously described. But there is no reference to the previous description. Even if there was a reference here, there is no mention of thermocycling conditions, nor is there any mention of how relative gene expression values are calculated. This is not acceptable methods reporting for qRT-PCR data.
Response 20: Thank you for your comment. The relevant details on quantitative real-time PCR analysis methods have been improved in section 4.8. Furthermore, detailed information has been provided in sections 4.2 and 4.3, and sections 4.3 and 4.4 have been integrated to assist readers in comprehending our research methodology. The pertinent modifications are reflected in lines 730-743.
Reviewer 2 Report
Comments and Suggestions for Authors
Comments are shown as speech baloons in the text (MS) file. This file is attached to be viewed by the authors.
Apart from the comments and some unclear parts of the text -which are shown in the attached file- the work is well written and the whole idea of the research is well described.
I applaud the authors for this work and I urge them to abandon the speculations for the future work to be done on the subject.

Author Response
Comment 1: Control of population size is meant through sex ratio manipulation? Could you explain it better?
Response 1: Thank you for your comment. The structure of the sentence has been subjected to rigorous analysis and subsequently revised in order to facilitate a clear introduction to the topic. The pertinent modifications are reflected in lines 42-47.
-
Comment 2: When first reference to the nematode Caenorhabditis elegans is made then the genus name should be written in full.
Response 2: Thank you for your comment. The requisite corrections have been implemented. The pertinent modifications are reflected in lines 53.
Comment 3: Add the prefix "the crab species" before "Eriocheir sinensis" and the prefix "the eastern river paws" before "Macrobrachium rosenbergii".
Response 3: Thank you for your comment. The requisite corrections have been implemented. The pertinent modifications are reflected in lines 63, 66.
Comment 4: The last sentence of section 2.4: “The expression of Dafem-1A and Dafem-1C genes were significantly higher in females than in males, while the expression levels of Dafem-1B genes were significantly lower in females than in males” something went wrong with the expression in this sentence.
Response 4: Thank you for your comment. The wording of the sentence has been amended in accordance with the requisite corrections. The pertinent modifications are reflected in lines 257-259.
Comment 5: What is meant by 'CK' in Figure 5?
Response 5: Thank you for your comment. The CK in Figure 5 is not treated with any terpenoids. We have added explanations about CK in Figure Legend.
Comment 6: What is meant by 'CK' in Figure 6?
Response 6: Thank you for your comment. The control group in Figure 6 was injected with an equal amount of diethyl pyrocarbonate (DEPC) water. We have added explanations about CK in Figure Legend.
Comment 7: What is meant by 'CK' in Figure 8?
Response 7: Thank you for your comment. The control group depicted in Figure 8 comprises insects that have not undergone RNAi treatment. Further clarification regarding CK has been provided in the figure legend.
Comment 8: In Discussion, “This is evidenced by a significant surplus of males and females. Furthermore, during the overwintering period, the body mass of males and females is significantly higher.”, does it refer to each developmental stage?
Response 8: Thank you for your comment. The male and female pronouns in this sentence represent the adult stage, and the sentence will be optimised in order to facilitate a more comprehensive understanding on the part of the reader. The pertinent modifications are reflected in lines 421-423.
Comment 9: Line 369-396. Though reasonably done it is a speculation.
Response 9: Thank you for your comment. We have condensed and refined this section to express our outlook on the practical application of this research. The pertinent modifications are reflected in lines 556-570.
Round 2
Reviewer 1 Report
Comments and Suggestions for Authors
The manuscript is substantially improved with greater clarity and reporting of more complete methodologies. However, further revisions are required before the manuscript will be ready for publication.
Introduction.
Line 44-47. “By silencing its sex determination gene through RNAi, impaired individuals can be generated reproductively and released them into the wild to mate with wild-type individuals, thereby producing sterile offspring and reducing their population size.”
In order for this claim to be made, there needs to be evidence that RNAi knockdown will result in sterility. In any case, this kind of statement seems more appropriate for the Discussion section, not the Introduction, which should provide a foundation in established literature for the research questions being addressed in the current study. The Discussion is more appropriate for forward looking aspirations.
Line 79. “insect-like effects”
What is meant by insect-like effects? Is it meant to say “effects in insects?”
Line 76-88. Paragraph on RNAi
This paragraph still reads too much like a general literature review. Has RNAi been explored and used in Dendroctonus species before? Has it been used -in any species- to knockdown fem1 genes? This kind of information is far more relevant to the current study than some text on delivery methods.
Line 99-100. “RNAi was employed to silence the Dafem-1 gene”
This is incorrect. It wasn’t one gene. Multiple Dafem-1 genes were targeted and silenced. There are other instances later on in the manuscript where mention is made of the or one Dafem-1 gene. Please ensure throughout the manuscript that these mentions are specific to multiple Dafem-1 genes, where appropriate.
Results.
Lines 174-181. Text on adult Dafem1-B expression in males and females.
The statements made in this section do not match the data plotted in Figure 2H. It seems like the text describing expression in males has been mixed up with the text describing expression in females. For example “expression of the male Dafem-1B gene was observed to be at a low level at both -10C and 20C. However, that is not the pattern observed in Figure 2H, which shows a statistically highest level of expression in males at 20C. Instead, the description here, lows at -10C and 20C -is- what is seen in females for Dafem-1B.
Line 211. “observed in rats”
What does this study have to do with rats?
Line 247. “the negative control treatment”
What is the negative control, specifically? It should be clearly indicated to leave no doubts or confusion.
Line 274-275. “while Dafem-1B and Dafem-1C decreased the most under +-3-carene treatment. (Figure 5B/C)” This statement is incorrect according to the data shown in Figures 5B and 5C. Please evaluate the data and correct the statement.
Line 275-280. Several statements made about which compounds correlated with the greatest inductions of each Dafem-1 gene in each of the different life stages. However, several of the statements made do not match up with the statistical data in each of the panels of Figure 5. For example, line 275-276, “In pupae, Dafem-1A gene expression decreased the most under -B-pinene treatment. However, according to panel 5D, there is statistically no difference in the decrease after -B-pinene, +B-pinene, and Limone treatments. There are several more instances like this. Please read carefully through the text and ensure the statements made match the statistically evaluated data in Figure 5.
Discussion.
Line 353. “cloned and identified”.
Logically, this should be “identified and cloned”
Line 366-367. “sex determination of creature”
This is grammatically incorrect. It might be better to say “living organisms” or something similar.
Line 398. “the Dafem-1 gene” should be “genes”
Line 406-407. “This suggests that the intake of nutrients is beneficial for maintaining active life activities, which in turn leads to an increase in gene expression.”
I think reality is more complex than this. Maintenance of life activities may involve increased expression of some genes, but also decreased expression of other genes.
Line 427-428. “during the developmental process of the newly emerged adults”
What does this mean? Do newly emerged adults undergo a developmental process?
Line 431-434. “This result is consistent with the gene expression levels at different developmental stages previously mentioned, indicating significant differences in gene express levels between males and females under different temperature, nutrition, feeding and terpenoid treatments.”
How can the result be consistent with gene expression levels at different developmental stages if males and females were not examined at different developmental stages, but rather only in adults. I think it would be more succinct to state that under all experimental conditions examined, the expression patterns were the same. 1A and 1C were more highly expressed in Females and 1B was more highly expressed in males.
Furthermore, the second half of this statement “under different temperature nutrition, feeding and terpenoid treatments” is repetitive to what was already mentioned on lines 426-428.
Line 471-473. “Following interference with Dafem-1A and Dafem-1C, the majority of females died, while the majority of males survived. Conversely, following interference with Dafem-1B, the majority of males died, while the majority of females survived, resulting in a significant imbalance in the ratio of females to males.”
These are stated as facts, but there is no direct evidence that males or females died. And other hypotheses could explain the distortion of sex ratios after gene knockdown, such as embryonic feminization or masculinization. It must be rewritten carefully to clarify that under the stated hypothesis, the females or males would die as a result of gene knockdown.
Line 496-498. “If we can obtain sterile mutants by RNAi and release them into the wild, so that the mutants will be mated with normal insects to produce steile progeny or lethal progeny.”
This needs to be rephrased. It is an incomplete thought, with the logical structure of: “If we do X, so that Y happens”. But the consequences of this are missing. What is the benefit of producing sterile or lethal progeny. It needs to be explicitly stated, to provide the “why” statement, as in why this is the ultimate goal of this work.
Materials and Methods.
Line 560. “Sequence analyses of Dafem-1”
This seems like it shouldn’t be here, or should be integrated with the next paragraph.
Line 563. “phylogenetic inference analysis of 7 full length sequences was performed using the neighbor-joining method with MEGA11.”
Which sequences were brought in for comparison? It should be mentioned here, not only just in the supplemental data file, and what type of alignment was done as input for the phylogenetic tree generation? These need to be mentioned with indication of which software was used and what parameters.
Line 590. “protein and cellulose reducing sugar”
This looks like a typo. It seems like it should be “protein, cellulose, and reducing sugar”
Figures and Tables.
Figure 5 legend. “The control group was not subjected to any terpenoid treatment.”
Please indicate what data in the figures is the control group. In other words, explicitly mention in the legend what CK means as you do for every other treatment on the X-axis. Because it is not clear what CK means without any definition of these letters.
Supplemental Figure S4. The figure legend says “The amplification of the Dafem-1 gene was analyzed using gel electrophoresis.” On line 106, it says “Three fem-1 genes were cloned and obtained in this experiment (Figure S4).
However, the amplicon sizes for Figure S4 do not look to match ORF sizes for Dafem-1A/B/C. the ladder indicates bands around 500bp, but full length ORFs are around 2200bp. Is the gel instead showing dsRNA products, or partial fragment ORFs confirming expression? This needs to be clarified.
Supplemental Figure S7. Based on the text Line 247-249, it is indicated that this is a negative control time-elapse plot for the feeding experiment. This is interpreted to mean the insects are not feeding for the data shown in this Figure, S7. Is that correct? If so, it must be noted that the X-axis title for each of the panels in S7 say “feeding duration”. However, if the insects are not feeding, then this is incorrect. It should be revised to say something like “time duration.”
Also, the title of the figure legend says “Relative expression of Dafem-1 genes in larvae and adults of D. armandi” with no reference to what experimental conditions are being examined. There needs to be a more descriptive title indicating that this is a time-elapse control for the feeding experiments, if that is the case.
Comments on the Quality of English LanguageThe quality of the English language is satisfactory, with a few minor edits needed, as indicated in the comments above.
Author Response
Response to Reviewers’ Comments
Dear Reviewer:
Thank you very much for your serious and detailed feedback and comments on my manuscript entitled “Fem-1 gene of Chinese white pine beetle (Dendroctonus armandi) function and response to environmental treatments” (ijms-3135279). This will be of great help in improving my manuscript and subsequent research. I have provided reasonable explanations and detailed revisions for all the above comments. I hope you can reconsider my manuscript. Here are my revision suggestions:
Comments to the Author:
The manuscript is substantially improved with greater clarity and reporting of more complete methodologies. However, further revisions are required before the manuscript will be ready for publication.
We appreciate for Reviewer’s warm work earnestly and hope that the correction will meet with approval.
Comments 1: Line 44-47. “By silencing its sex determination gene through RNAi, impaired individuals can be generated reproductively and released them into the wild to mate with wild-type individuals, thereby producing sterile offspring and reducing their population size.” In order for this claim to be made, there needs to be evidence that RNAi knockdown will result in sterility. In any case, this kind of statement seems more appropriate for the Discussion section, not the Introduction, which should provide a foundation in established literature for the research questions being addressed in the current study. The Discussion is more appropriate for forward looking aspirations.
Response 1: Thank you for your comment. The statement in question has been removed, and the relevant changes are reflected in lines 44-47. Furthermore, lines 514-519 express our wishes regarding the aforementioned sentence.
Comments 2: Line 79. “insect-like effects”. What is meant by insect-like effects? Is it meant to say “effects in insects?”
Response 2: Thank you for your comment. The relevant sentences have been revised to indicate that the inaugural successful application of RNAi to insects was in the species Drosophila melanogaster. The requisite amendments are reflected in lines 78-79.
Comments 3: Line 76-88. Paragraph on RNAi. This paragraph still reads too much like a general literature review. Has RNAi been explored and used in Dendroctonus species before? Has it been used -in any species- to knockdown fem1 genes? This kind of information is far more relevant to the current study than some text on delivery methods.
Response 3: Thank you for your comment. We have recognised the limitations of this section in relation to RNAi, and as the content described does not relate to our research, we would like to provide an overview of RNAi as a method. We have adapted our description and made revisions based on your comment. The relevant changes are reflected in line 84-92.
Comments 4: Line 99-100. “RNAi was employed to silence the Dafem-1 gene”. This is incorrect. It wasn’t one gene. Multiple Dafem-1 genes were targeted and silenced. There are other instances later on in the manuscript where mention is made of the or one Dafem-1 gene. Please ensure throughout the manuscript that these mentions are specific to multiple Dafem-1 genes, where appropriate.
Response 4: Thank you for your comment. We have revised the description of this section in the manuscript to indicate that our study focuses on three Dafem-1 genes to avoid inaccurate description. The relevant changes are reflected in line 104, 127, 349, 359, 404, 408, 410, 417, 435-436, 442, 453, 501, 505, 506, 514, 529, 618 and 648.
Comments 5: Lines 174-181. Text on adult Dafem1-B expression in males and females. The statements made in this section do not match the data plotted in Figure 2H. It seems like the text describing expression in males has been mixed up with the text describing expression in females. For example “expression of the male Dafem-1B gene was observed to be at a low level at both -10C and 20C. However, that is not the pattern observed in Figure 2H, which shows a statistically highest level of expression in males at 20C. Instead, the description here, lows at -10C and 20C -is- what is seen in females for Dafem-1B.
Response 5: Thank you for your comment. We would like to express our gratitude for your invaluable support of our research. The wording in this section has been checked and corrected in order to avoid significant errors in the description. The requisite amendments are reflected in lines 179–187.
Comments 6: Line 211. “observed in rats”. What does this study have to do with rats?
Response 6: Thank you for your comment. The erroneous statement has been corrected in order to avoid any significant errors in the description. The requisite amendments are reflected in line 217.
Comments 7: Line 247. “the negative control treatment”. What is the negative control, specifically? It should be clearly indicated to leave no doubts or confusion.
Response 7: Thank you for your comment. We have paid close attention to the rigor of these results and have provided more detailed explanations in order to eliminate any ambiguity or vagueness. The requisite amendments are reflected in line 253-256.
Comments 8: Line 274-275. “while Dafem-1B and Dafem-1C decreased the most under +-3-carene treatment. (Figure 5B/C)” This statement is incorrect according to the data shown in Figures 5B and 5C. Please evaluate the data and correct the statement. Line 275-280. Several statements made about which compounds correlated with the greatest inductions of each Dafem-1 gene in each of the different life stages. However, several of the statements made do not match up with the statistical data in each of the panels of Figure 5. For example, line 275-276, “In pupae, Dafem-1A gene expression decreased the most under -B-pinene treatment. However, according to panel 5D, there is statistically no difference in the decrease after -B-pinene, +B-pinene, and Limone treatments. There are several more instances like this. Please read carefully through the text and ensure the statements made match the statistically evaluated data in Figure 5.
Response 8: Thank you for your comment. In order to more accurately convey the nature of our research, section 2.6 has been rewritten. The requisite amendments are reflected in line 278-307.
Comments 9: Line 353. “cloned and identified”. Logically, this should be “identified and cloned”
Response 9: Thank you for your comment. The wording has been amended in accordance with the requisite corrections. The requisite amendments are reflected in line 372.
Comments 10: Line 366-367. “sex determination of creature”. This is grammatically incorrect. It might be better to say “living organisms” or something similar.
Response 10: Thank you for your comment. On a single occasion, the term “feature” was replaced with a more appropriate term, namely “living organisms”. The requisite amendments are reflected in line 386.
Comments 11: Line 398. “the Dafem-1 gene” should be “genes”
Response 11: Thank you for your comment. The grammatical errors identified in this section have been rectified, and the same corrections have been applied to other sections of the manuscript. The requisite amendments are reflected in line 104, 127, 349, 359, 404, 408, 410, 417, 435-436, 442, 453, 501, 505, 506, 514, 529, 618 and 648.
Comments 12: Line 406-407. “This suggests that the intake of nutrients is beneficial for maintaining active life activities, which in turn leads to an increase in gene expression.”. I think reality is more complex than this. Maintenance of life activities may involve increased expression of some genes, but also decreased expression of other genes.
Response 12: Thank you for your comment. It has been observed that the wording in question is not precise. Consequently, it has been corrected to read “changes in gene expression levels”. The requisite amendments are reflected in line 425-426.
Comments 13: Line 427-428. “during the developmental process of the newly emerged adults”. What does this mean? Do newly emerged adults undergo a developmental process?
Response 13: Thank you for your comment. It has been observed that the aforementioned statement is erroneous. To obviate any potential confusion, it is necessary to provide an accurate representation of the facts. The intention is to convey that there are notable disparities in the gene expression profiles of male and female Dafem-1 genes in response to various stimuli, including temperature, nutrition, feeding, and terpenoid compound treatment. The requisite amendments are reflected in line 446-.
Comments 14: Line 431-434. “This result is consistent with the gene expression levels at different developmental stages previously mentioned, indicating significant differences in gene express levels between males and females under different temperature, nutrition, feeding and terpenoid treatments.”. How can the result be consistent with gene expression levels at different developmental stages if males and females were not examined at different developmental stages, but rather only in adults. I think it would be more succinct to state that under all experimental conditions examined, the expression patterns were the same. 1A and 1C were more highly expressed in Females and 1B was more highly expressed in males. Furthermore, the second half of this statement “under different temperature nutrition, feeding and terpenoid treatments” is repetitive to what was already mentioned on lines 426-428.
Response 14: Thank you for your comment. We concur with your assessment and have implemented the requisite modifications to the wording of the sentence in question, as well as the removal of redundant expressions, in accordance with your feedback. The requisite amendments are reflected in line 448-451.
Comments 15: Line 471-473. “Following interference with Dafem-1A and Dafem-1C, the majority of females died, while the majority of males survived. Conversely, following interference with Dafem-1B, the majority of males died, while the majority of females survived, resulting in a significant imbalance in the ratio of females to males.”. These are stated as facts, but there is no direct evidence that males or females died. And other hypotheses could explain the distortion of sex ratios after gene knockdown, such as embryonic feminization or masculinization. It must be rewritten carefully to clarify that under the stated hypothesis, the females or males would die as a result of gene knockdown
Response 15: Thank you for your comment. The expression of this sentence has been subjected to careful consideration and has been the subject of extensive discussion based on the ideas you have provided. Furthermore, relevant literature has been consulted in order to support our findings. Furthermore, while the assertion of male and female mortality is, at this stage, merely a conjecture, the lack of rigour in its formulation is also a cause for concern. Accordingly, the sentence in question has been subjected to a rigorous process of revision and rewriting. The requisite amendments are reflected in line 486-492.
Comments 16: Line 496-498. “If we can obtain sterile mutants by RNAi and release them into the wild, so that the mutants will be mated with normal insects to produce steile progeny or lethal progeny.”. This needs to be rephrased. It is an incomplete thought, with the logical structure of: “If we do X, so that Y happens”. But the consequences of this are missing. What is the benefit of producing sterile or lethal progeny. It needs to be explicitly stated, to provide the “why” statement, as in why this is the ultimate goal of this work.
Response 16: Thank you for your comment. We have realized the incompleteness of this sentence and have supplemented the missing parts. The requisite amendments are reflected in line 514-519.
Comments 17: Line 560. “Sequence analyses of Dafem-1”. This seems like it shouldn’t be here, or should be integrated with the next paragraph.
Response 17: Thank you for your comment. The formatting errors have been rectified. The requisite amendments are reflected in line 581.
Comments 18: Line 563. “phylogenetic inference analysis of 7 full length sequences was performed using the neighbor-joining method with MEGA11.” Which sequences were brought in for comparison? It should be mentioned here, not only just in the supplemental data file, and what type of alignment was done as input for the phylogenetic tree generation? These need to be mentioned with indication of which software was used and what parameters.
Response 18: Thank you for your comment. The majority of the content in section 4.4 has been rewritten with the aim of providing a more precise and detailed account of the specific methodology employed in the construction of evolutionary trees utilising multiple sequences. The requisite amendments are reflected in line 583-597.
Comments 19: Line 590. “protein and cellulose reducing sugar”. This looks like a typo. It seems like it should be “protein, cellulose, and reducing sugar”
Response 19: Thank you for your comment. The aforementioned erroneous statement has been duly rectified. The requisite amendments are reflected in line 622.
Comments 20: Figure 5 legend. “The control group was not subjected to any terpenoid treatment.”. Please indicate what data in the figures is the control group. In other words, explicitly mention in the legend what CK means as you do for every other treatment on the X-axis. Because it is not clear what CK means without any definition of these letters.
Response 20: Thank you for your comment. The designation "CK" is used to denote a "control check." For purposes of clarity, we have incorporated detailed information in the figure legend. Furthermore, additional modifications have been made throughout the text, including the introduction of the term 'CK' in Figure 8. The requisite amendments are reflected in line 311, 313, 359, 363.
Comments 21: Supplemental Figure S4. The figure legend says “The amplification of the Dafem-1 gene was analyzed using gel electrophoresis.” On line 106, it says “Three fem-1 genes were cloned and obtained in this experiment (Figure S4). However, the amplicon sizes for Figure S4 do not look to match ORF sizes for Dafem-1A/B/C. the ladder indicates bands around 500bp, but full length ORFs are around 2200bp. Is the gel instead showing dsRNA products, or partial fragment ORFs confirming expression? This needs to be clarified.
Response 21: Thank you for your comment. Figure S4 depicts a portion of the target gene fragments of the three Dafem-1 genes, rather than the entirety of the gene sequences. In order to provide a more detailed description, the manuscript and supplementary materials have been revised. The requisite amendments are reflected in line 110-111 of manuscript, figure legend of figure S4.
Comments 22: Supplemental Figure S7. Based on the text Line 247-249, it is indicated that this is a negative control time-elapse plot for the feeding experiment. This is interpreted to mean the insects are not feeding for the data shown in this Figure, S7. Is that correct? If so, it must be noted that the X-axis title for each of the panels in S7 say “feeding duration”. However, if the insects are not feeding, then this is incorrect. It should be revised to say something like “time duration.”. Also, the title of the figure legend says “Relative expression of Dafem-1 genes in larvae and adults of D. armandi” with no reference to what experimental conditions are being examined. There needs to be a more descriptive title indicating that this is a time-elapse control for the feeding experiments, if that is the case.
Response 22: Thank you for your comment. The objective of Figure S7 is to eliminate the potential impact of variables such as age, duration, and aging on the feeding duration treatment. This is a negative control set-up for the feeding duration treatment experiment. To facilitate the interpretation of our experimental results, we have made the requisite modifications to the content of Figure S7.